# RestoRect: Degraded Image Restoration via Latent Rectified Flow & Feature Distillation

## Abstract

Current approaches for restoration of degraded images face a critical trade-off: high-performance models are too slow for practical use, while fast models produce poor results. Knowledge distillation transfers teacher knowledge to students, but existing static feature matching methods cannot capture how modern transformer architectures dynamically generate features. We propose **'RestoRect'**, a novel Latent Rectified Flow Feature Distillation method for restoring degraded images. We apply rectified flow to reformulate feature distillation as a generative process where students learn to synthesize teacher-quality features through learnable trajectories in latent space. Our framework combines Retinex theory for physics-based decomposition with learnable anisotropic diffusion constraints, and trigonometric color space polarization. We introduce a Feature Layer Extraction loss for robust knowledge transfer between different network architectures through cross-normalized transformer feature alignment with percentile-based outlier detection. RestoRect achieves better training stability, and faster convergence and inference while preserving restoration quality. We demonstrate superior results across 15 image restoration datasets, covering 4 tasks, on 10 metrics.

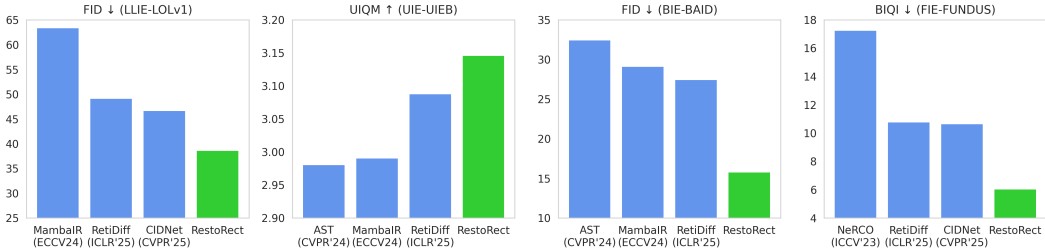

Figure 1: RestoRect achieves superior performance on four image restoration tasks.

## 1 Introduction

Image restoration from degraded inputs including low-light (LLIE), underwater (UIE), backlit (BAID), and fundus (FIE) enhancement, remains a key challenge in computer vision. Real-world images often suffer from illumination degradation, noise, and compression artifacts that impair both human perception and downstream tasks. Traditional optimization-based methods exploit physical priors but falter on images with complex degradations, while transformer-based deep learning achieves strong restoration by learning rich multi-scale features. Generative approaches further enhance quality, with diffusion models operating in latent spaces and integrating Retinex priors to capture the complex distributions of natural images. However, such gains incur steep computational costs, limiting real-time use. Knowledge distillation offers efficiency by transferring knowledge from large teachers to compact students, but struggles with transformer-based restoration. Conventional approaches compute static feature losses between teacher and student layers, neglecting the dynamic feature generation of multi-head attention and layer interactions. This mismatch hampers dependency modeling, degrading student performance. Recent models such as Reti-Diff (He et al., 2023) (Retinex priors) and HVI-CIDNet (Yan et al., 2024) (learnable color spaces) achieve good restoration, but their distillation relies on static feature matching, which fails to capture generative processes.

We propose **RestoRect**, which formulates knowledge distillation as a generative process through latent rectified flow. Instead of matching static features, student networks learn the dynamic synthesis of features through flow matching dynamics, using linear interpolation trajectories in latent space between noise and target features. This reduces sampling steps while preserving feature quality.

At the core of RestoRect is the **F**eature **L**ayer **EX**traction (FLEX) Loss, designed to address distribution mismatch in feature distillation. Unlike prior methods that assume teacher and student features share the same statistical space, FLEX normalizes both using student statistics, enabling meaningful comparison despite evolving feature distributions during training. To further stabilize learning, percentile-based outlier detection mitigates noisy or corrupted regions. Our framework integrates classical image processing with modern generative modeling: Retinex theory for physics-based decomposition, learnable anisotropic diffusion for structural consistency, and trigonometric color space polarization to eliminate the red discontinuity artifacts common in image transformations. Together, these components preserve both texture and color in restored images.

RestoRect employs a two-stage training paradigm for feature distillation. In Stage 1, the teacher network is trained with pixel, perceptual, and physics-based losses to achieve high-quality restoration. Stage 2 distills knowledge into the student via latent rectified flow. In its first phase, only rectified flow velocity predictors are trained while the main restoration network remains frozen. The pre-trained teacher extracts high-quality Retinex and image features from paired degraded and ground-truth inputs, which serve as targets for two rectified flow models. These models learn velocity fields that reproduce teacher-level features through learnable trajectories, enabling synthesis in only a few steps. In the second phase, the full restoration network is trained using these generative processes: velocity predictors dynamically generate student features, which are aligned with teacher features via our FLEX Loss that cross-normalizes multi-scale transformer representations and applies percentile-based outlier detection. This design allows the student to efficiently learn and generate teacher-quality features, achieving restoration performance comparable to diffusion-based methods while operating at significantly higher efficiency.

Our key technical contributions include: **1.** A novel framework modeling knowledge transfer as a generative process using latent rectified flow, where the student network learns velocity fields to synthesize teacher-quality features. **2.** A novel U-Net transformer architecture with Spatial Channel Layer Normalization (SCLN) and Query-Key normalization, for attention stability under degraded inputs. **3.** A novel Feature Layer EXtraction (FLEX) Loss using feature statistics to normalize both teacher and student representations for multi-scale alignment in transformers. **4.** Combining Retinex theory with learnable anisotropic diffusion constraints and trigonometric color space polarization to eliminate artifacts and boost restoration quality.

## 2 RELATED WORK

**Degraded Image Restoration** has evolved from classical signal processing to modern deep learning frameworks. Early approaches such as histogram equalization (Cheng & Shi, 2004), gamma correction (Huang et al., 2012), and Retinex theory (Edwin, 1977) provided interpretable solutions but failed to generalize across degradations. Retinex-based extensions (Fu et al., 2016; Li et al., 2018) incorporated physical priors for reflectance–illumination decomposition, yet remained constrained by hand-crafted assumptions. Deep learning enabled data-driven feature learning, with convolutional models by (Wei et al., 2018) and by (Wang et al., 2019) leveraging Retinex decomposition for improved color correction. Transformer-based methods further enhanced global illumination consistency (Zamir et al., 2022), while adaptive designs by (Xu et al., 2022) and state space models like by (Guo et al., 2024) advanced efficiency and context modeling. Specialized solutions addressed low-light enhancement (Guo et al., 2020; Jiang et al., 2021), underwater restoration (Naik et al., 2021; Guo et al., 2023), and backlit enhancement (Gaintseva et al., 2024; Jiang et al., 2021). Hybrid approaches such as by (He et al., 2025b) bridged optimization- and learning-based paradigms via deep unfolding, while by (Yan et al., 2024; 2025) introduced learnable color-space transformations to decouple brightness and chromaticity.

**Image Generative Modeling** aims to capture complex data distributions and synthesize realistic details. GAN-based methods (Cong et al., 2023; Jiang et al., 2021) achieved high-quality results but suffered from instability and mode collapse. Diffusion models improved fidelity through iterative denoising (Yi et al., 2023), though efficiency remained limited. Latent-space diffusion, such as Reti-

Diff (He et al., 2023), reduced overhead by incorporating Retinex priors. Flow-based approaches offered exact likelihoods and stable training (Kingma & Dhariwal, 2018), with rectified flow (Liu et al., 2022) enabling efficient straight-line sampling. Integrating generative priors into restoration networks has driven advances in knowledge distillation (Hinton et al., 2015), conditional and multi-scale generation (Saharia et al., 2022; Ho et al., 2022), and physics-informed restoration (Xia et al., 2023). Nonetheless, achieving real-time, high-fidelity restoration remains challenging due to the trade-off between generative quality and computational efficiency.

**Knowledge Distillation** enables compact models to inherit capabilities from larger teachers (Hinton et al., 2015). Early methods matched intermediate features (Romero et al., 2014) or attention maps (Zagoruyko & Komodakis, 2016), using L2 losses (Heo et al., 2019) or attention transfer (Huang & Wang, 2017). For vision transformers, challenges from multi-head attention and positional encodings inspired approaches like distillation tokens in DeiT (Touvron et al., 2021) and attention matrix alignment (Wang et al., 2020). However, these strategies treat features as static targets, overlooking the dynamic generation in transformer architectures (Jiao et al., 2019). In image restoration, distillation is further complicated by multi-scale feature dependencies and complex distributions (Zhang et al., 2022; Berrada et al., 2025). Architectural mismatches between teacher and student amplify these gaps, limiting transfer efficiency and degrading restoration quality, motivating new paradigms that model feature generation as a learnable process rather than static matching (Bing et al., 2025).

## 3 METHODOLOGY

### 3.1 PROBLEM FORMULATION

We tackle efficient knowledge distillation for degraded image restoration, aiming to transfer knowledge from a powerful teacher $\mathcal{F}_T$ to a lightweight student $\mathcal{F}_S$ without sacrificing quality. Given a degraded input $I_{LQ} \in \mathbb{R}^{H \times W \times 3}$ and ground truth $I_{GT} \in \mathbb{R}^{H \times W \times 3}$, the objective is: $\mathcal{F}_S(I_{LQ}) \approx \mathcal{F}_T(I_{LQ}) \approx I_{GT}$. The main challenge is feature distribution mismatch between teacher and student. Standard distillation aligns features with simple distance metrics, which breaks down when distributions differ significantly, especially in transformer-based networks where multi-head attention produces features with varying means, variances, and outlier characteristics.

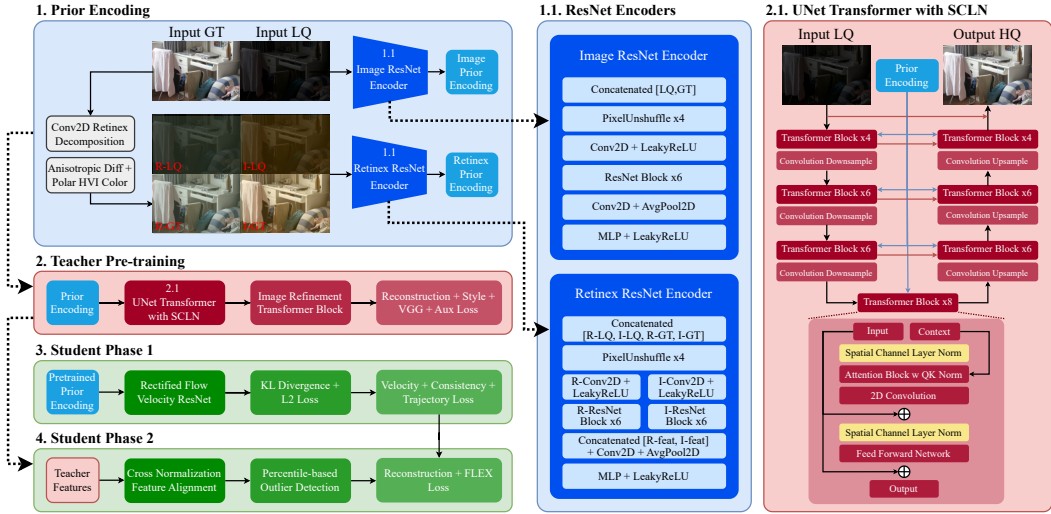

Figure 2: Training framework flowchart for RestoRect. Starting from top left (1. Prior Encoding) the inputs go through retinex decomposition and pass through encoders (1.1 ResNet Encoders) to prepare image and retinex prior encodings. Next these prior encodings are pre-trained (2. Teacher Pre-training) with the teacher model (2.2 UNet Transformer with SCLN) using a reconstruction loss. Finally the frozen prior encodings and teacher model are used for student phase 1 and phase 2 training using rectified flow loss. Full architecture details in Appendix A.7.

## 3.2 TEACHER NETWORK TRANSFORMER PRETRAINING

Our method uses well-established Retinex theory to derive physics-informed features as priors for knowledge distillation. Retinex models an image $I$ as the product of reflectance $R$ and illumination $L$: $I = R \odot L$, where $R$ encodes surface properties and $L$ captures lighting. We use two decomposition networks, $\mathcal{D}_l$ (low-light) and $\mathcal{D}_h$ (normal-light), each mapping $\mathcal{D}(I) \to (R, L)$ with $R \in \mathbb{R}^{H \times W \times 3}$ and $L \in \mathbb{R}^{H \times W \times 1}$ (Wu et al., 2022; He et al., 2023). This dual setup ensures robust decomposition under diverse lighting. The decomposed components are then encoded (Figure 2(1)): a Retinex encoder extracts features from $[R; L]$ via reflectance (192-dim) and illumination (64-dim) pathways, while an image encoder processes raw image features to preserve holistic appearance. Our teacher network uses U-Net transformer architecture (Huang et al., 2020; Cao et al., 2022) with key innovations for robust image restoration. The hierarchical transformer architecture processes multi-scale representations through encoder-decoder structures with skip connections, incorporating specialized normalization and attention mechanisms designed for degraded image inputs. Traditional layer normalization operates independently on spatial and channel dimensions, potentially losing critical spatial correlations essential for restoration tasks.

**Spatial Channel Layer Normalization (SCLN)** is introduced that captures global image statistics: $\text{SCLN}(x) = (x - \mu_{global}) / (\sqrt{\sigma_{global}^2 + \epsilon}) \cdot \gamma$, where the global statistics are computed across flattened spatial-channel dimensions. This novel formulation ensures that normalization captures both local spatial patterns and global image characteristics, with learnable channel-wise scaling $\gamma \in \mathbb{R}^C$ that adapts to different feature semantics. Transformer-based restoration suffers from attention instability during training, particularly with degraded inputs which have irregular noise patterns and missing information. We apply normalization to query and key representations before attention computation, which prevents attention weight saturation in degraded regions, and ensures stable gradients throughout the attention mechanism: $\text{Attn}(Q, K, V) = \text{softmax}\left(\frac{\text{Norm}(Q) \cdot \text{Norm}(K)^T}{\sqrt{d_k}} \cdot \tau\right) V$. The teacher network processes both raw images and their Retinex decompositions through separate pathways. This design allows queries from reflectance components to attend to illumination structure, preserving intrinsic scene properties. Figure 3 shows in blue our SCLN with QK norm achieves more stable training compared to vanilla layer normalization without QK norm in red. To our knowledge no previous restoration method has used this transformer architecture.

Table 1: Computational overhead comparison between LayerNorm (LN) and Spatial Channel Layer Norm (SCLN) across different precisions on an MLP network. Results averaged over 3 random seeds on 512×512×64 resolution.

| | Single Layer Performance (ms) | | | | Multi-layer Performance (ms) | | | |
|---|---|---|---|---|---|---|---|---|
| Precision | LayerNorm | SCLN | Overhead | (%) | LayerNorm | SCLN | Overhead | (%) |
| FP32 | $0.7059 \pm 0.0004$ | $0.7113 \pm 0.0000$ | 0.0054 | +0.76 | $3.05 \pm 0.00$ | $3.06 \pm 0.00$ | 0.01 | +0.33 |
| FP16 | $0.1899 \pm 0.0000$ | $0.1991 \pm 0.0001$ | 0.0092 | +4.82 | $1.21 \pm 0.00$ | $1.17 \pm 0.00$ | −0.05 | −3.72 |
| BF16 | $0.1935 \pm 0.0001$ | $0.2004 \pm 0.0001$ | 0.0069 | +3.57 | $1.23 \pm 0.00$ | $1.17 \pm 0.00$ | −0.06 | −4.67 |

The benchmark results in Table 1 show that SCLN introduces minimal computational overhead due to its efficient normalization strategy. At the single-layer level, SCLN incurs only 0.76% overhead in FP32, as computing statistics across spatial-channel dimensions requires marginally more operations than channel-wise normalization. The slightly higher percentage overhead in FP16/BF16 (4.82% and 3.57%) is primarily an artifact of the dramatically reduced absolute inference times, LayerNorm executes so quickly in lower precision that even negligible absolute differences appear larger percentagewise. At the full network level, SCLN's overhead becomes negligible (0.33% in FP32) or even negative (-3.72% in FP16, -4.67% in BF16), suggesting superior memory access patterns and cache efficiency when operations are repeated across multiple layers. Standard LayerNorm's repeated reshape operations (to_3d/to_4d conversions) accumulate overhead, while SCLN's direct 4D tensor operations benefit from better spatial locality and reduced memory bandwidth pressure. This explains why SCLN actually becomes faster than LayerNorm in lower-precision full-network scenarios, making the trade-off highly favorable with PSNR improvement with no speed penalty. Note that we train our RestoRect models with FP32, and Table 1 is a toy MLP example.

**Auxiliary Constraints** like anisotropic diffusion (Perona et al., 1994) and polarized HVI color spaces (Yan et al., 2024) (Yan et al., 2025) are incorporated that enforce edge-preserving texture matching and eliminate artifacts. The anisotropic diffusion operator computes: $\mathcal{A}(I) = \nabla \cdot (c(|\nabla I|)\nabla I)$, with the diffusion coefficient defined as: $c(|\nabla I|) = \exp\left(-|\nabla I|^2/s^2\right)$, where $s$ is a learnable sensitivity parameter initialized as $s = 0.1$ and constrained to $s \in [0.01, 1.0]$ to prevent numerical instability. The texture consistency loss enforces structural similarity between input and predicted reflectance: $L_{tex} = \|\mathcal{A}(I_{input}) - \mathcal{A}(R_{pred})\|_1$. This constraint preserves essential edge structures while suppressing noise, maintains texture coherence across different scales, and provides gradient-based supervision for fine-grained details. We additionally enforce illumination smoothness through gradient-aware weighting: $L_{lum} = \sum_{i,j} w_{i,j} \left(|\nabla_x L_{i,j}|^2 + |\nabla_y L_{i,j}|^2\right)$, where $w_{i,j} = \exp(-|\nabla L_{i,j}|)$ provides adaptive regularization based on local gradient magnitude. Standard image color spaces exhibit critical limitations for restoration like discontinuities at the red boundary ($H = 0°$ and $H = 360°$) and degenerate mappings in dark regions. To address these fundamental limitations, polarized HVI (Horizontal-Vertical-Intensity) color space is introduced that eliminates these artifacts through trigonometric parameterization. The polarized transformation maps hue to continuous coordinates: $H_{polar} = C_k \cdot S \cdot \cos(\pi H/3), V_{polar} = C_k \cdot S \cdot \sin(\pi H/3), I_{polar} = I_{max} = \max(R, G, B)$, where the adaptive intensity collapse factor is: $C_k = k \cdot \sin(\pi I_{max}/2) + \epsilon$, with learnable density parameter $k$ initialized to 1.0 and constrained to $k \in [0.1, 5.0]$. This formulation eliminates red discontinuity through periodic parameterization, provides robustness through adaptive intensity collapse that prevents degenerate mappings in dark regions, and maintains color relationships under illumination changes. While (Yan et al., 2024) (Yan et al., 2025) frames HVI as a representation transformation, we define an explicit color loss in HVI space. The polarized color loss is computed as:

$$L_{col} = \|H_{polar}^{pred} - H_{polar}^{gt}\|_1 + \|V_{polar}^{pred} - V_{polar}^{gt}\|_1 + \|I_{polar}^{pred} - I_{polar}^{gt}\|_1 \qquad (1)$$

The primary reconstruction objective employs pixel-wise supervision through L1 loss: $L_{rec} = \|I_{pred} - I_{gt}\|_1$, where $I_{pred}$ represents the network's restored output and $I_{gt}$ denotes the ground truth high-quality image. To capture perceptual similarity beyond pixel-level differences, we incorporate perceptual loss using pre-trained VGG (Simonyan & Zisserman, 2014)features. The perceptual loss extracts multi-scale feature representations that align with human visual perception: $L_{vgg} = \sum_l \lambda_l \|\phi_l(I_{pred}) - \phi_l(I_{gt})\|_2^2$, where $\phi_l$ represents VGG features at layer $l$, and $\lambda_l$ denotes layer-specific weights that emphasize semantically important features. We additionally incorporate style loss that captures texture and artistic consistency through Gram matrix matching: $L_{sty} = \sum_l \|G_l(\phi_l(I_{pred})) - G_l(\phi_l(I_{gt}))\|_F^2$, where $G_l(\phi_l(I)) = \phi_l(I)\phi_l(I)^T$ computes the Gram matrix at layer $l$, and $\|\cdot\|_F$ denotes the Frobenius norm. This novel combination ensures that the restored images maintain both structural accuracy and perceptual realism. The complete teacher training objective combines these losses:

$$L_{teach} = L_{rec} + L_{vgg} + L_{sty} + \lambda_{tex}L_{tex} + \lambda_{col}L_{col} + \lambda_{lum}L_{lum} \qquad (2)$$

with $\lambda_{tex} = 0.05$, $\lambda_{col} = 0.05$, and $\lambda_{lum} = 0.2$. Figure 3 shows in green our how our auxiliary constraints allow training of a stronger teacher model with faster convergence. (He et al., 2023) previously used reconstruction and style loss with perceptual VGG features. To our knowledge, we are the first to implement anisotropic diffusion texture and illumination smoothness constraints with explicit HVI color loss.

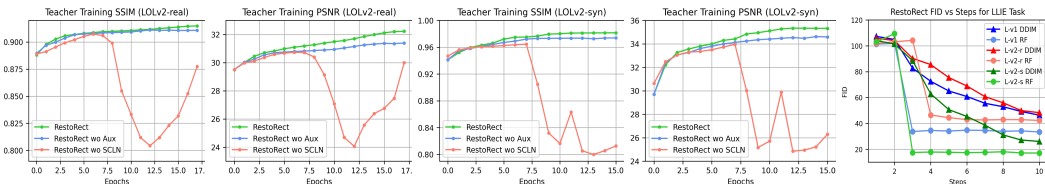

Figure 3: (1-4) Teacher model training with ablations of SCLN & QK Norm (red) and auxiliary losses (blue). (5) FID vs Steps inference performance show Rectified Flow (RF) student model producing high quality images in fewer steps compared to Denoising Diffusion Implicit Model (DDIM).

### 3.3 STUDENT NETWORK TRAINING WITH LATENT RECTIFIED FLOW

Traditional knowledge distillation treats feature transfer as static matching between teacher and student representations. This approach suffers from several limitations including assuming compatible feature distributions between architectures, lacking flexibility in handling multi-modal feature distributions, and being unable to adapt to varying complexity of restoration tasks. We reformulate knowledge distillation as a generative process using rectified flow, which models feature synthesis through straight-line paths in latent space. Given teacher features $\mathbf{f}_{teach} \in \mathbb{R}^d$ and noise $\mathbf{z} \sim \mathcal{N}(0, I)$, rectified flow defines the interpolation path: $\mathbf{x}_t = (1 - t)\mathbf{z} + t\mathbf{f}_{teach}, \quad t \in [0, 1]$. The velocity field represents the direction of optimal transport: $\mathbf{v}(\mathbf{x}_t, t) = \frac{d\mathbf{x}_t}{dt} = \mathbf{f}_{teach} - \mathbf{z}$. We train separate velocity prediction networks $\epsilon_\theta^{rex}$ and $\epsilon_\theta^{img}$ for reflectance and image features using the velocity matching objective: $L_{vel} = \mathbb{E}_{t,\mathbf{z},\mathbf{f}_{teach}} \left[ \|\epsilon_\theta(\mathbf{x}_t, t, \mathbf{c}) - \mathbf{v}(\mathbf{x}_t, t)\|_2^2 \right]$, where $\mathbf{c}$ represents conditioning information from the input image. Each velocity predictor implements a Residual MLP architecture. During inference, we solve the ODE using Euler's method with adaptive step sizing: $\mathbf{x}_{t+\Delta t} = \mathbf{x}_t + \Delta t \cdot \epsilon_\theta(\mathbf{x}_t, t, \mathbf{c})$. This requires only 1-4 integration steps compared to 10+ steps for DDIM models, providing significant computational advantages. Standard knowledge distillation losses (KL divergence, L2 distance) assume that teacher and student features exist in compatible statistical distributions. This assumption fails for complex transformer architectures, and when fine-tuning on different datasets, leading to suboptimal knowledge transfer (Lin et al., 2022).

**FLEX (Feature Layer EXtraction) Loss** addresses feature distribution mismatch through cross-normalization for distribution alignment, percentile-based outlier detection for robust training, and dynamic resolution-aware weighting for multi-scale importance. Unlike (Berrada et al., 2025) which is specialized for diffusion autoencoders, FLEX provides a general-purpose distillation loss that transfers feature distributions across heterogeneous teacher-student architectures. The key method is cross-normalization using student statistics. For each layer $l$, FLEX normalizes both teacher and student features using student statistics:

$$\mu_{\text{stud}}^l = \text{mean}(\mathbf{f}_{\text{stud}}^l), \quad \sigma_{\text{stud}}^l = \text{std}(\mathbf{f}_{\text{stud}}^l) + \epsilon, \quad \mathbf{f}_{\text{teach}}^{l,\text{norm}} = \frac{\mathbf{f}_{\text{teach}}^l - \mu_{\text{stud}}^l}{\sigma_{\text{stud}}^l}, \quad \mathbf{f}_{\text{stud}}^{l,\text{norm}} = \frac{\mathbf{f}_{\text{stud}}^l - \mu_{\text{stud}}^l}{\sigma_{\text{stud}}^l}$$

This aligns both features to the student's distribution, enabling meaningful comparison across architecture capacity differences. FLEX incorporates fast percentile-based outlier detection to handle extreme values that destabilize training. This masking strategy prioritizes training stability over complete spatial coverage, as extreme outliers generate destabilizing gradients that outweigh their informational value. The outlier mask identifies reliable spatial locations: $M_{\text{reliable}}^{l,c,h,w} = \mathbb{I}[|\mathbf{f}_{\text{stud}}^{l,c,\text{norm},h,w}| \leq \tau_p^{l,c}]$, where $\tau_p^{l,c}$ is the p-th percentile of normalized feature magnitudes for layer l, channel c, with p=95% by default. FLEX computes dynamic resolution-based weights:

$$w_l^{\text{res}} = \max \left( (H_{\text{base}} W_{\text{base}} / H_l W_l)^{0.25}, 0.1 \right)$$

where $(H_{\text{base}}, W_{\text{base}}) = (64, 64)$ ensures appropriate weighting across resolutions. The complete FLEX loss combines masked feature matching with dual weighting:

$$L_{\text{FLEX}} = \sum_l w_l^{\text{layer}} \cdot w_l^{\text{res}} \cdot \frac{\sum_{c,h,w} M_{\text{reliable}}^{l,c,h,w} \cdot \|\mathbf{f}_{\text{teach}}^{l,c,\text{norm},h,w} - \mathbf{f}_{\text{stud}}^{l,c,\text{norm},h,w}\|^2}{\sum_{c,h,w} M_{\text{reliable}}^{l,c,h,w} + \epsilon} \tag{3}$$

where $w_l^{\text{layer}}$ represents predefined layer weights and the denominator normalizes by reliable elements. FLEX includes SNR-aware application, activating only when $t/T < \tau_{\text{SNR}} = 0.4$, focusing distillation on cleaner intermediate states. Cross-normalization enables stable transfer between different architectures, outlier detection prevents training instability, dynamic weighting balances multi-scale contributions, and streaming processing optimizes memory usage. Standard KD methods lack these capabilities, assuming compatible distributions and uniform spatial weighting.

**Trajectory Consistency Regularization** is introduced to ensure smooth and semantically consistent rectified flow trajectories, which prevents erratic feature generation and maintains coherence throughout the ODE integration process (Yang et al., 2024). We enforce smooth transitions between consecutive ODE steps: $L_{trans} = \sum_{i=1}^{N-1} \|\mathbf{f}_{pred}^{i+1} - \mathbf{f}_{pred}^i\|_2^2$, where $\mathbf{f}_{pred}^i$ represents predicted features at the $i$-th integration step. We ensure final generated features align with teacher targets: $L_{target} = \|\mathbf{f}_{pred}^{final} - \mathbf{f}_{teach}\|_2^2$. We enforce consistency in semantic feature representations across

the trajectory: $L_{cons} = \sum_{i=1}^{N} \text{cos\_dist}(\mathbf{f}_{pred}^{i}, \mathbf{f}_{teach})$. The complete trajectory consistency loss is: $L_{traj} = \alpha_{trans} L_{trans} + \alpha_{target} L_{target} + \alpha_{cons} L_{cons}$, with $\alpha_{trans} = 0.1$, $\alpha_{target} = 0.5$, and $\alpha_{cons} = 0.2$. Our training protocol addresses the challenge of jointly learning velocity prediction and restoration quality through a principled two-phase approach. We first train rectified flow components while freezing the main restoration network:

$$L_{phase1} = L_{vel}^{rex} + L_{vel}^{img} + \lambda_{KD} L_{KD} + \lambda_{traj} L_{traj} \tag{4}$$

This phase establishes stable velocity prediction capabilities without interference from restoration objective gradients. We use separate optimizers for reflectance and image velocity predictors with learning rates $lr_{rex} = 2 \times 10^{-4}$ and $lr_{img} = 2 \times 10^{-4}$. The complete network is then trained using features generated by learned velocity predictors, where $\lambda_{FLEX} = 0.15$, $\lambda_{vel} = 0.05$:

$$L_{phase2} = L_{rec} + \lambda_{FLEX} L_{FLEX} + \lambda_{vel}(L_{vel}^{rex} + L_{vel}^{img}) \tag{5}$$

## 4 EXPERIMENTS

**Experimental Setup.** We implement our model in PyTorch and trained it on 8 NVIDIA H100 GPUs. Teacher pretraining is performed for 15-20 epochs depending on dataset convergence, while student phases I and II are each trained for 10 epochs. We use Adam optimizer with momentum terms (0.9, 0.999). For fair comparison with prior work (He et al., 2023), we adopt the same configuration of transformer blocks, attention heads, and channel dimensions: [3, 3, 3, 3], [1, 2, 4, 8], and [64, 128, 256, 512] from levels 1-4. During inference, we make 4 function evaluation calls for rectified flow generation, yielding faster generation and higher-quality outputs compared to state-of-the-art methods. Training follows the methodology of Reti-Diff and CIDNet across datasets and tasks.

**Quantitative Evaluation.** For the low-light image enhancement (LLIE) task, we conduct experiments on LOL-v1 (Wei et al., 2018), LOL-v2-real, LOL-v2-syn (Yang et al., 2021), and SID (Chen et al., 2019). Performance is evaluated with PSNR, SSIM, FID, and BIQI (Hore & Ziou, 2010; Moorthy & Bovik, 2010), where higher PSNR/SSIM and lower FID/BIQI indicate better results. RestoRect achieves state-of-the-art performance across all datasets shown in Table 2, with improvements on almost every metric over the second-best methods (RetiDiff and CIDNet). The visual results shown in Figure 4 highlight clear improvements in fine grained details shown in cyan boxes (please zoom in for clarity).

For the underwater image enhancement (UIE) task, we evaluate on UIEB (Li et al., 2019) and LSUI (Peng et al., 2023), using PSNR, SSIM, and UIQM (Panetta et al., 2015). Higher values across all metrics indicate better performance. RestoRect outperforms RetiDiff by 1.76dB PSNR on UIEB and matches its performance on LSUI while achieving superior SSIM scores shown in Table 3. For the backlit image enhancement (BAID) task, experiments are performed (Lv et al., 2022), with evaluation on PSNR, SSIM, and FID. RestoRect demonstrates substantial improvements with 4.48dB PSNR gain over RetiDiff and 11.65 FID reduction shown in Table 4. Additionally, we test on real-world fundus image enhancement (FIE) (Shen et al., 2020) images using the LOL-v2-syn pretrained model, evaluating with BIQI and CLIPIQA (Wang et al., 2023b), where higher CLIPIQA values indicate better performance. RestoRect achieves the lowest BIQI score of 6.033, outperforming SNRNet shown in Table 5. The visual results shown in Figure 5 highlight our performance with details shown in yellow boxes (please zoom in for clarity). We note that Reti-Diff baseline images for UIEB and LSUI in middle row very closely match the ground truth while the scores are marginally worse than ours. This makes us believe that the publicly available checkpoints provided on Reti-Diff's github might be overfitted to the validation set, unlike our model which has not seen the validation set.

For real-world image restoration, we test on five unpaired datasets: DCIM (Lee et al., 2013), LIME (Guo et al., 2016), MEF (Wang et al., 2013), NPE (Ma et al., 2015), and VV (He et al., 2025a). Using the LOL-v2-syn pretrained model for inference, we evaluate with BRISQUE (Mittal et al., 2012), where lower values are better. RestoRect consistently outperforms CIDNet across most datasets, achieving the best scores on DCIM (16.56), LIME (16.12), and VV (24.42) as shown in Table 6. We further evaluate on single image contrast enhancement (SICE) (Cai et al., 2018), which contains underexposed and overexposed images, training on the resized SICE training set and test on the datasets SICE-Mix and SICE-Grad (Zheng et al., 2022) with metrics PSNR, SSIM, LPIPS.

Table 2: LLIE task results. Best result shown in Green and second best shown in Blue. We also include our performance on RestoRect reconstruction teacher model.

| Methods | LOL-v1 | | | | LOL-v2-real | | | | LOL-v2-syn | | | | SID | | | |
|---|---|---|---|---|---|---|---|---|---|---|---|---|---|---|---|---|
| | PSNR↑ | SSIM↑ | FID↓ | BIQI↓ | PSNR↑ | SSIM↑ | FID↓ | BIQI↓ | PSNR↑ | SSIM↑ | FID↓ | BIQI↓ | PSNR↑ | SSIM↑ | FID↓ | BIQI↓ |
| MRQ (Liu et al., 2023) | 25.24 | 0.855 | 53.32 | 22.73 | 22.37 | 0.854 | 68.89 | 33.61 | 25.54 | 0.940 | 21.56 | 25.09 | 24.80 | 0.688 | 63.72 | 29.53 |
| IAGC (Wang et al., 2023c) | 24.53 | 0.866 | 59.73 | 25.50 | 22.20 | 0.863 | 70.34 | 31.70 | 25.58 | 0.941 | 21.58 | 30.32 | 23.17 | 0.640 | 78.80 | 30.56 |
| DiffIR (Xia et al., 2023) | 23.15 | 0.828 | 70.13 | 26.38 | 21.15 | 0.816 | 72.33 | 29.15 | 24.76 | 0.921 | 21.36 | 27.74 | 23.17 | 0.640 | 78.80 | 30.56 |
| CUE (Zheng et al., 2023) | 21.86 | 0.841 | 69.83 | 27.15 | 21.19 | 0.829 | 67.05 | 28.83 | 24.41 | 0.917 | 31.34 | 33.83 | 23.25 | 0.652 | 77.38 | 28.85 |
| GSAD (Hou et al., 2023) | 20.33 | 0.852 | 51.64 | 19.96 | 20.90 | 0.847 | 46.77 | 28.85 | 24.22 | 0.927 | 19.24 | 25.76 | – | – | – | – |
| AST (Zhou et al., 2024) | 21.09 | 0.858 | 87.67 | 21.23 | 21.68 | 0.857 | 91.81 | 25.17 | 22.25 | 0.927 | 19.20 | 20.78 | – | – | – | – |
| Mamba (Guo et al., 2024) | 22.33 | 0.863 | 63.39 | 20.17 | 21.97 | 0.840 | 56.09 | 24.46 | 25.75 | 0.958 | 17.95 | 20.37 | 21.14 | 0.656 | 154.76 | 32.72 |
| RetiDiff (He et al., 2023) | 25.35 | 0.866 | 49.14 | 17.75 | 22.97 | 0.858 | 43.18 | 23.66 | 27.53 | 0.951 | 13.82 | 15.77 | 25.53 | 0.692 | 51.66 | 25.58 |
| CIDNet (Yan et al., 2024) | 23.50 | 0.900 | 46.69 | 14.77 | 24.11 | 0.871 | 48.04 | 18.45 | 25.71 | 0.942 | 18.60 | 15.87 | 22.90 | 0.676 | 55.29 | 29.12 |
| RestoRect (teacher only) | 22.18 | 0.862 | 63.77 | 26.50 | 20.11 | 0.833 | 65.84 | 29.21 | 23.15 | 0.911 | 28.72 | 28.13 | 22.60 | 0.717 | 68.42 | 27.13 |
| RestoRect | 27.84 | 0.945 | 38.67 | 8.35 | 22.97 | 0.911 | 42.80 | 10.47 | 27.69 | 0.968 | 16.75 | 11.67 | 26.19 | 0.923 | 54.23 | 19.57 |

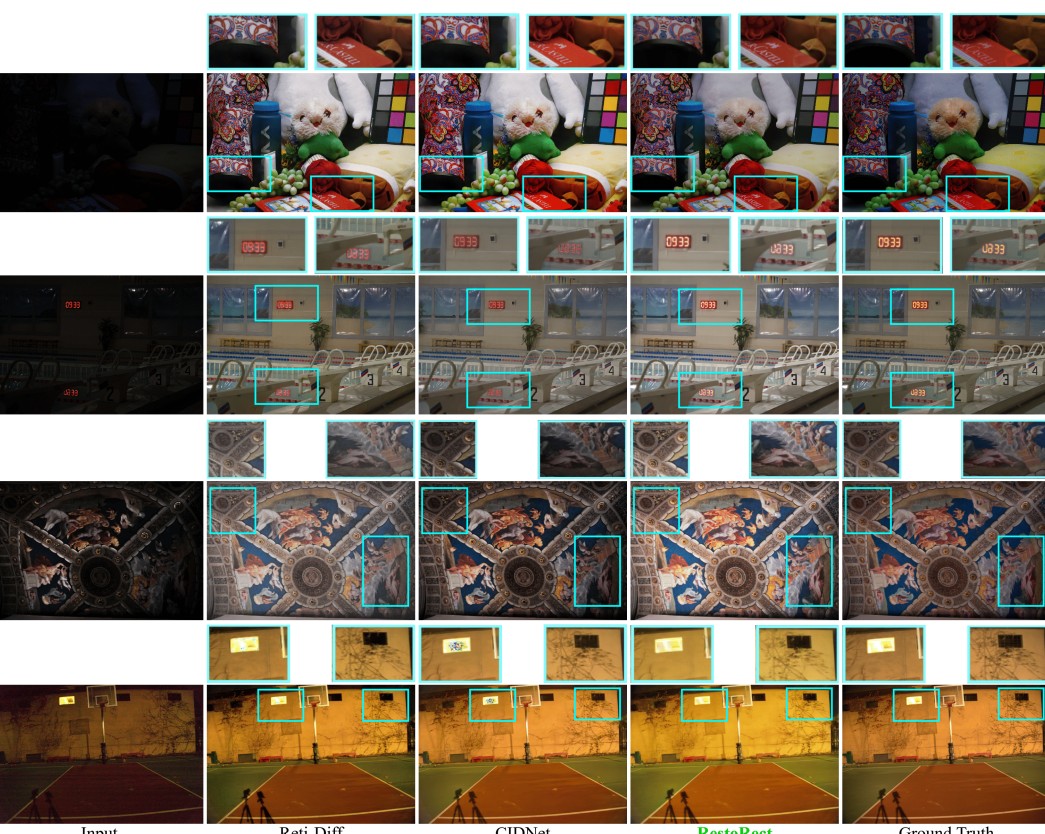

Figure 4: LLIE task visual results (Top to Bottom: LOL-v1, v2-real, v2-syn, SID). Clear improvements in fine grained details are shown in cyan boxes (please zoom in for clarity).

RestoRect achieves superior PSNR and SSIM performance over CIDNet by 1.6dB and 0.031 on SICE-Mix, and 2.0dB and 0.077 on SICE-Grad, as shown in Table 7.

**Qualitative Evaluation.** We conduct a user study to evaluate low-light image enhancement. Eight participants are shown 20 low-light images alongside enhanced outputs from RestoRect, Reti-Diff, and CIDNet (RAVE included for BAID dataset). In a blind comparison, subjects are asked to select the result that appears closest to the ground truth. Figure 8 presents the preference distributions, showing that RestoRect consistently achieves the highest preference across all five datasets, highlighting its ability to generate visually appealing results perceived as closest to the ground truth. Figure 8 and Table 11 shows comparison of RestoRect's student model parameter size (M) and GFLOPs against other transformer architecture baselines demonstrating efficiency.

Table 3: UIE task results

| Methods | UIEB | | | LSUI | | |
|---|---|---|---|---|---|---|
| | PSNR↑ | SSIM↑ | UIQM↑ | PSNR↑ | SSIM↑ | UIQM↑ |
| SUnwet (Naik et al., 2021) | 18.28 | 0.855 | 2.942 | 20.89 | 0.875 | 2.746 |
| PUIE (Fu et al., 2022) | 21.38 | 0.882 | 3.021 | 23.70 | 0.902 | 2.974 |
| UShape (Peng et al., 2023) | 22.91 | 0.905 | 2.896 | 24.16 | 0.917 | 3.022 |
| PUGAN (Cong et al., 2023) | 23.05 | 0.897 | 2.902 | 25.06 | 0.916 | 3.106 |
| ADP (Zhou et al., 2023) | 22.90 | 0.892 | 3.005 | 24.28 | 0.913 | 3.075 |
| NU2Net (Guo et al., 2023) | 22.38 | 0.903 | 2.936 | 25.07 | 0.908 | 3.112 |
| AST (Zhou et al., 2024) | 22.19 | 0.908 | 2.981 | 27.46 | 0.916 | 3.107 |
| Mamba (Guo et al., 2024) | 22.60 | 0.939 | 2.991 | 27.68 | 0.916 | 3.118 |
| RetiDiff (He et al., 2023) | 24.12 | 0.910 | 3.088 | 28.10 | 0.929 | 3.208 |
| RestoRect | 25.88 | 0.950 | 3.121 | 28.10 | 0.937 | 3.229 |

Table 4: BAID task results

| Methods | BAID | | |
|---|---|---|---|
| | PSNR↑ | SSIM↑ | FID↓ |
| EnGAN (Jiang et al., 2021) | 17.96 | 0.819 | 43.55 |
| URetinex (Wu et al., 2022) | 19.08 | 0.845 | 42.26 |
| CLIPLIT (Liang et al., 2023) | 21.13 | 0.853 | 37.30 |
| DiffRet (Yi et al., 2023) | 22.07 | 0.861 | 38.07 |
| DiffIR (Xia et al., 2023) | 21.10 | 0.835 | 40.35 |
| AST (Zhou et al., 2024) | 22.61 | 0.851 | 32.47 |
| Mamba (Guo et al., 2024) | 23.07 | 0.874 | 29.13 |
| RAVE (Gaintseva et al., 2024) | 21.26 | 0.872 | 64.89 |
| RetiDiff (He et al., 2023) | 23.19 | 0.876 | 27.47 |
| RestoRect | 27.67 | 0.965 | 15.82 |

Table 5: FIE task results

| Methods | Fundus | |
|---|---|---|
| | BIQI↓ | CLIPQ↑ |
| SNRNet (Xu et al., 2022) | 6.144 | 0.557 |
| URetinex (Wu et al., 2022) | 12.158 | 0.561 |
| SCI (Ma et al., 2022) | 23.527 | 0.552 |
| MIRNet (Zamir et al., 2022) | 14.925 | 0.527 |
| FourLLE (Wang et al., 2023a) | 7.741 | 0.508 |
| CUE (Zheng et al., 2023) | 11.721 | 0.448 |
| NeRCO (Yang et al., 2023) | 17.256 | 0.451 |
| RetiDiff (He et al., 2023) | 10.788 | 0.525 |
| CIDNet (Yan et al., 2024) | 10.663 | 0.529 |
| RestoRect | 6.033 | 0.503 |

Figure 5: FIE (Top), UIEB (Middle Left), LSUI (Middle Right), BAID (Bottom) task visual results. Clear improvements in fine grained details are shown in yellow boxes (please zoom in for clarity).

**Ablation and Generalizability.** Figure 3 presents the results of teacher model training under different ablation settings. The removal of auxiliary constraints, such as anisotropic diffusion and the polarized HVI color space loss, is shown in blue. In contrast, the ablation of SCLN and QK normalization from the transformer block is shown in red, where a standard layer normalization and vanilla QK computation are used instead, following (He et al., 2023). As illustrated in green, the teacher model achieves the best performance with RestoRect when all proposed components are included. Table 8 further reports student model performance across different training and testing conditions on the LOL-v1, LOL-v2-real, and LOL-v2-synthetic datasets. In the table, '-FLEX' denotes models trained on the same dataset as the test set but without the FLEX loss. The FLEX training strategy demonstrates substantial improvements, with gains across all metrics compared to the full model results shown in Table 2. Subsequent rows in Table 8 evaluate cross-dataset transfer, where models trained on one dataset are tested on another, highlighting their strong generalization capacity. These results demonstrate that models trained for a given task can effectively transfer knowledge and serve as strong initialization points for fine-tuning on other datasets. Figure 7 shows visual results for LLIE task with ablation of SCLN and QK Norm, Auxiliary constraints, FLEX loss, compared to full RestoRect architecture and Ground Truth. Figure 3 also demonstrates the FID performance of RestoRect across different inference steps for the LLIE task. Our rectified flow formulation consistently outperforms (He et al., 2023) DDIM across all LLIE datasets, generating restored image within 3-4 steps, making it ideal for real time applications.

Table 6: Unpaired task results

| Methods | DCIM | LIME | MEF | NPE | VV |
|---|---|---|---|---|---|
| | BRISQUE↓ | | | | |
| KinD (Zhang et al., 2019) | 48.72 | 39.91 | 49.94 | 36.85 | 50.56 |
| ZeroDCE (Guo et al., 2020) | 27.56 | 20.44 | 17.32 | 24.72 | 34.66 |
| RUAS (Liu et al., 2021) | 38.75 | 27.59 | 23.68 | 47.85 | 38.37 |
| LLFlow (Wang et al., 2022) | 26.36 | 27.06 | 30.27 | 28.86 | 31.67 |
| SNRAware (Xu et al., 2022) | 37.35 | 39.22 | 31.28 | 26.65 | 78.72 |
| PairLIE (Fu et al., 2023) | 33.31 | 25.23 | 27.53 | 28.27 | 39.13 |
| CIDNet (Yan et al., 2024) | 21.47 | 16.25 | 13.77 | 18.92 | 30.63 |
| RestoRect | 16.56 | 16.12 | 14.69 | 23.91 | 24.42 |

Table 7: SICE task results

| Methods | SICE-Mix | | | SICE-Grad | | |
|---|---|---|---|---|---|---|
| | PSNR↑ | SSIM↑ | LPIPS↓ | PSNR↑ | SSIM↑ | LPIPS↓ |
| RetiNet (Wei et al., 2018) | 12.397 | 0.606 | 0.407 | 12.450 | 0.619 | 0.364 |
| ZeroDCE (Guo et al., 2020) | 12.428 | 0.633 | 0.382 | 12.475 | 0.644 | 0.334 |
| URetinex (Wu et al., 2022) | 10.903 | 0.600 | 0.402 | 10.894 | 0.610 | 0.356 |
| RUAS (Liu et al., 2021) | 8.684 | 0.493 | 0.525 | 8.628 | 0.494 | 0.499 |
| LLFlow (Wang et al., 2022) | 12.737 | 0.617 | 0.388 | 12.737 | 0.617 | 0.388 |
| LEDNet (Zhou et al., 2022) | 12.668 | 0.579 | 0.412 | 12.551 | 0.576 | 0.383 |
| CIDNet (Yan et al., 2024) | 13.425 | 0.636 | 0.362 | 13.446 | 0.648 | 0.318 |
| RestoRect | 15.041 | 0.667 | 0.393 | 15.447 | 0.715 | 0.354 |

Table 8: Ablation

| Test | Train | PSNR↑ | SSIM↑ | FID↓ | BIQI↓ |
|---|---|---|---|---|---|
| v1 | -FLEX | 24.27 | 0.891 | 44.75 | 9.02 |
| | v2-s | 18.32 | 0.827 | 99.36 | 18.74 |
| | v2-r | 17.57 | 0.827 | 111.66 | 21.68 |
| v2-r | -FLEX | 23.16 | 0.880 | 41.55 | 10.52 |
| | v1 | 22.27 | 0.874 | 48.92 | 18.57 |
| | v2-s | 21.15 | 0.837 | 106.29 | 22.92 |
| v2-s | -FLEX | 27.89 | 0.942 | 17.93 | 11.95 |
| | v1 | 19.96 | 0.876 | 69.37 | 16.39 |
| | v2-r | 17.18 | 0.768 | 117.84 | 25.26 |

Figure 6: DCIM (Row 1 Left), LIME (Row 1 Right), MEF (Row 2 Left), NPE (Row 2 Right), VV (Row 3), SICE-Grad (Row 4) task visual results. Clear improvements in fine grained details are shown in red boxes (please zoom in for clarity).

The student model ablation study on LOL-v1 dataset in Table 9 evaluates the contribution of each component in the knowledge distillation framework. The metrics demonstrate that the combination of Rectified Flow (RF) with Trajectory Consistency (RF+TC) and FLEX loss components yields substantial improvements over traditional KD methods of (He et al., 2023), with the full configuration achieving 27.84 PSNR, 0.945 SSIM, 38.67 FID, and 8.35 BIQI on LOL-v1. The incremental addition of FLEX components shows that Cross-Normalization (CN) alone provides modest gains, while adding Percentile Masking (PM) delivers significant improvements (27.82 PSNR), and the complete FLEX formulation with Resolution Weighting (RW) achieves optimal performance. Hyperparameter analysis reveals that the percentile threshold of 95% for outlier detection and SNR threshold of 0.4 provide the best balance between training stability and performance.

**Failure Cases.** Despite strong performance across many restoration tasks, RestoRect is limitated in extreme degradation scenarios seen in Figure 9. Failure occurs when input images contain severe overexposure or underexposure with complete information loss in large spatial regions. The model produces unrealistic artifacts including color bleeding, checkerboard-like noise patterns, and hallucinatory textures that deviate from natural image statistics. These artifacts are highlighted in the

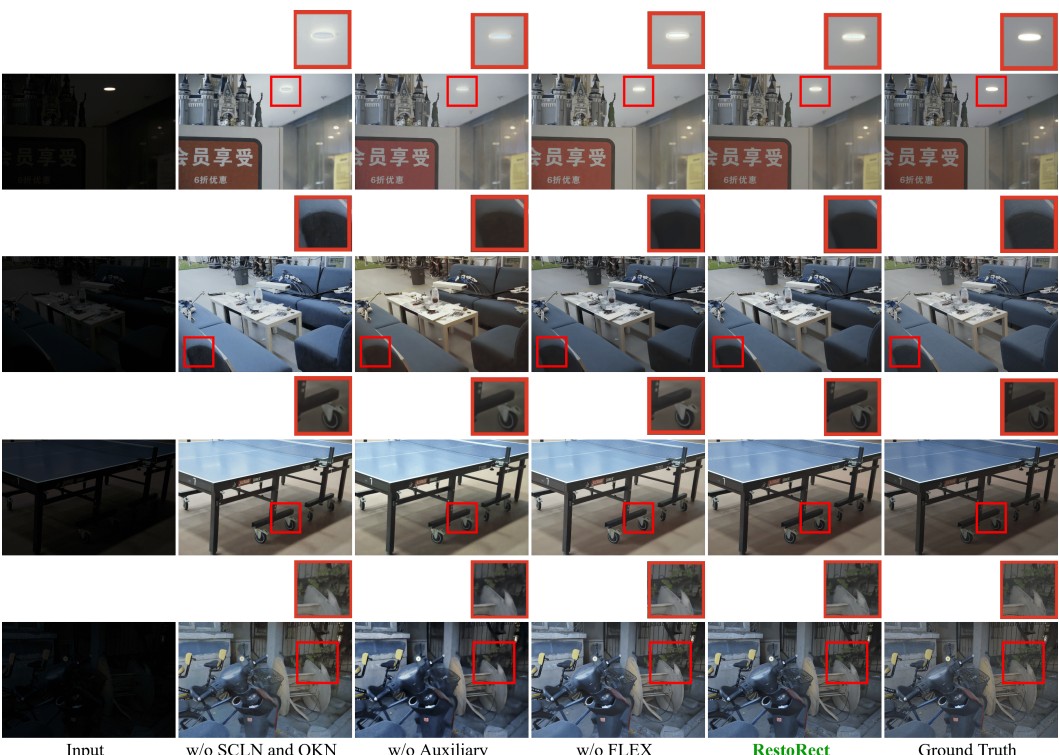

Input | w/o SCLN and QKN | w/o Auxiliary | w/o FLEX | **RestoRect** | Ground Truth

Figure 7: LLIE task visual results with ablation of SCLN and QK Norm, Auxiliary constraints, FLEX loss, compared to full RestoRect model and Ground Truth.

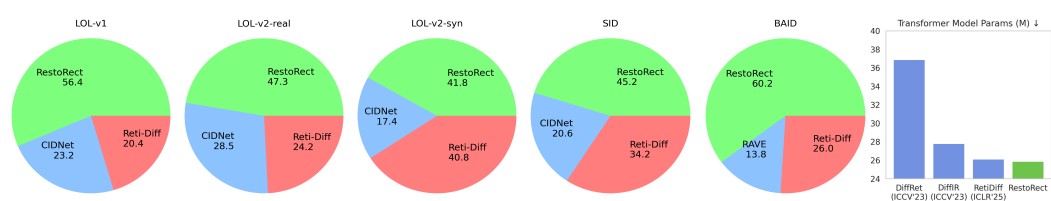

Figure 8: Qualitative human evaluation user study on LLIE and BAID datasets. Student model parameter size (M) comparison against other transformer architecture baselines showing efficiency.

zoomed regions in red boxes, where recovered areas show synthetic-looking patterns rather than co-herent scene content. These cases reveal that when the signal-to-noise ratio falls below a threshold, velocity prediction networks struggle to generate meaningful features, resulting in the model relying on learned priors that introduce perceptually implausible reconstructions.

Table 9: Student ablation study (L-v1).

| Configuration | PSNR↑ | SSIM↑ | FID↓ | BIQI↓ |
|---|---|---|---|---|
| RetiDiff (Traditional KD) | 25.35 | 0.866 | 49.14 | 17.75 |
| RF(no TC) + no FLEX | 24.10 | 0.855 | 49.28 | 17.95 |
| RF(TC) + no FLEX | 24.27 | 0.891 | 44.75 | 9.02 |
| RF(TC) + FLEX(CN) | 25.64 | 0.913 | 42.18 | 8.73 |
| RF(TC) + FLEX(CN+PM) | 27.82 | **0.947** | 39.25 | 8.42 |
| RF(TC) + FLEX(CN+PM+RW) | **27.84** | 0.945 | **38.67** | **8.35** |
| Percentile Threshold $p = 85\%$ | 26.21 | 0.898 | 39.84 | 8.58 |
| Percentile Threshold $p = 90\%$ | 27.58 | **0.946** | 39.12 | 8.43 |
| Percentile Threshold $p = 95\%$ | **27.84** | 0.945 | **38.67** | **8.35** |
| Percentile Threshold $p = 99\%$ | 27.67 | 0.943 | 38.91 | 8.41 |
| SNR Threshold $\tau_{SNR} = 0.2$ | 27.15 | 0.937 | 40.12 | 8.64 |
| SNR Threshold $\tau_{SNR} = 0.4$ | 27.84 | **0.945** | **38.67** | **8.35** |
| SNR Threshold $\tau_{SNR} = 0.6$ | **27.89** | 0.941 | 39.28 | 8.47 |
| SNR Threshold $\tau_{SNR} = 0.8$ | 26.92 | 0.934 | 40.53 | 8.71 |

Figure 9: Failure cases on SICE-Mix dataset.

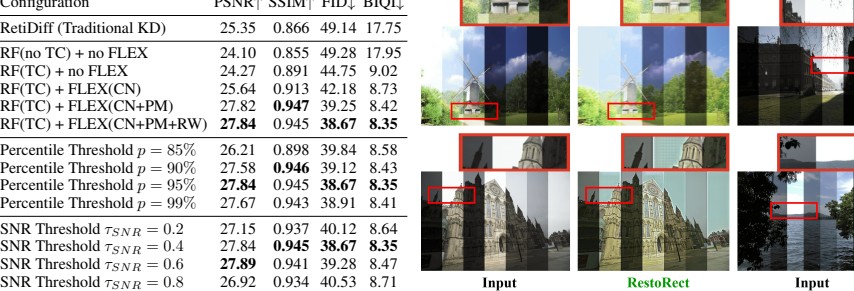

Input | **RestoRect** | Input | **RestoRect**

**Downstream Application.** For downstream application, RestoRect demonstrates strong transfer capability on the ExDark (Loh & Chan, 2019) low-light object detection task shown in Table 10. Following (Cai et al., 2023) and (He et al., 2023), low light images from the ExDark dataset were restored and object-detection task was performed using YOLOv3 model. Images enhanced by RestoRect achieve 77.1% mean average precision across 12 object categories, outperforming RetiDiff (He et al., 2023) (75.8%) and other methods. Improvements are observed in categories like Bicycle (85.8%), Bottle (79.5%), and Motor (77.9%), indicating that the restoration quality translates to real-world vision tasks.

Table 10: Downstream application on low-light image detection task on ExDark dataset.

| Methods | Bicycle | Boat | Bottle | Bus | Car | Cat | Chair | Cup | Dog | Motor | People | Table | Mean |
|---|---|---|---|---|---|---|---|---|---|---|---|---|---|
| Restormer | 77.0 | 71.0 | 68.8 | 91.6 | 77.1 | 62.5 | 57.3 | 68.0 | 69.6 | 69.2 | 74.6 | 49.7 | 69.7 |
| SCI | 73.4 | 68.0 | 69.5 | 86.2 | 74.5 | 63.1 | 59.5 | 61.0 | 67.3 | 63.9 | 73.2 | 47.3 | 67.2 |
| SNR-Net | 78.3 | 74.2 | 74.5 | 89.6 | 82.7 | 66.8 | 66.3 | 62.5 | 74.7 | 63.1 | 73.3 | 57.2 | 71.9 |
| Retformer | 78.1 | 74.5 | 74.2 | 91.2 | 82.2 | 65.0 | 63.3 | 67.0 | 75.4 | 68.6 | 75.3 | 55.6 | 72.5 |
| RetiDiff | 82.0 | 77.9 | 76.4 | 92.2 | 83.3 | 69.6 | 67.4 | 74.4 | 75.5 | 74.3 | 78.3 | 57.9 | 75.8 |
| RestoRect | 85.8 | 79.1 | 79.5 | 91.5 | 83.5 | 70.2 | 68.8 | 74.1 | 78.2 | 77.9 | 78.2 | 57.4 | 77.1 |

Table 11: Transformer model complexity comparison

| Methods | Source | Params M | GFLOPs |
|---|---|---|---|
| Restormer | CVPR'22 | 26.13 | 144.25 |
| Diff-Reti | ICCV'23 | 56.88 | 198.16 |
| DiffIR | ICCV'23 | 27.80 | 35.32 |
| GSAD | NIPS'23 | 17.17 | 670.33 |
| Reti-Diff | ICLR'25 | 26.11 | 87.63 |
| RestoRect | Ours | 25.87 | 49.50 |

Table 12: RestoRect Image Quality Evaluation Results for all 15 datasets across 10 metrics

| Dataset | PSNR | SSIM | FID | NIQE | LPIPS | BRISQ | BIQI | UCIQE | UIQM | CLIPQ |
|---|---|---|---|---|---|---|---|---|---|---|
| LOL-v1 | 27.85 | 0.94 | 38.67 | 7.47 | 0.11 | 27.16 | 8.35 | 0.52 | 2.60 | 0.499 |
| LOL-v2 Real | 22.97 | 0.91 | 42.81 | 7.74 | 0.13 | 28.44 | 10.48 | 0.51 | 2.88 | 0.500 |
| LOL-v2 Syn | 27.70 | 0.97 | 16.75 | 5.74 | 0.06 | 15.68 | 11.68 | 0.55 | 2.77 | 0.498 |
| SID | 26.19 | 0.92 | 54.23 | 5.87 | 0.15 | 20.05 | 19.57 | 0.85 | 2.38 | 0.498 |
| UIEB | 25.89 | 0.95 | 20.26 | 6.49 | 0.11 | 17.89 | 13.99 | 0.58 | 3.12 | 0.501 |
| LSUI | 28.10 | 0.94 | 17.83 | 5.04 | 0.18 | 21.82 | 16.06 | 0.57 | 3.23 | 0.499 |
| BAID | 27.68 | 0.97 | 15.83 | 8.11 | 0.06 | 34.39 | 10.49 | 0.56 | 2.87 | 0.501 |
| Fundus | 20.45 | 0.92 | 37.04 | 8.27 | 0.06 | 27.54 | 6.03 | 0.60 | 2.06 | 0.503 |
| DCIM | 19.92 | 0.82 | 72.72 | 6.36 | 0.17 | 16.57 | 10.26 | 0.57 | 2.33 | 0.499 |
| LIME | 18.36 | 0.76 | 101.31 | 6.12 | 0.21 | 16.13 | 11.76 | 0.59 | 2.19 | 0.497 |
| MEF | 17.20 | 0.69 | 74.06 | 6.13 | 0.26 | 14.70 | 11.23 | 0.56 | 2.83 | 0.499 |
| NPE | 16.28 | 0.77 | 63.75 | 7.10 | 0.18 | 23.91 | 12.91 | 0.53 | 2.64 | 0.498 |
| VV | 17.45 | 0.80 | 91.08 | 7.55 | 0.20 | 24.42 | 9.81 | 0.63 | 2.20 | 0.498 |
| SICE (mix) | 15.04 | 0.67 | 125.23 | 6.60 | 0.39 | 21.88 | 11.51 | 0.54 | 3.02 | 0.496 |
| SICE (grad) | 15.45 | 0.72 | 80.86 | 6.32 | 0.35 | 21.98 | 11.16 | 0.54 | 2.95 | 0.497 |

# 5 CONCLUSION

We present RestoRect, a generative knowledge distillation framework that reformulates degraded image restoration through latent rectified flow. Unlike traditional approaches that rely on static feature matching, RestoRect models feature transfer through learnable trajectories and introduces the FLEX loss for principled distribution alignment. Combined with a specialized U-Net transformer architecture and physics-based constraints, our method achieves state-of-the-art results across 15 datasets covering low-light, underwater, backlit, and fundus enhancement. RestoRect delivers better perceptual quality with only 4 inference steps, making it both effective and computationally efficient. Beyond restoration, this generative distillation method highlights new opportunities for efficient model compression and cross-architecture transfer in computer vision, establishing potential foundation for broader advances in fast high-quality image, and video restoration for future work.

# 6 LIST OF ACRONYMS

Table 13: List of acronyms used in this paper

| Acronym | Full Form |
| --- | --- |
| *Tasks & Methods* | |
| LLIE | Low-Light Image Enhancement |
| UIE | Underwater Image Enhancement |
| FIE | Fundus Image Enhancement |
| SCLN | Spatial Channel Layer Normalization |
| FLEX | Feature Layer EXtraction Loss |
| *Datasets* | |
| LOL | Low-light Outdoor Lighting |
| SID | See in the Dark |
| UIEB | Underwater Image Enhancement Bench |
| LSUI | Large Scale Underwater Image |
| BAID | Backlit Image Dataset |
| DCIM | Digital Camera Image |
| LIME | Low-light Image Enhancement |
| MEF | Multi-Exposure Fusion |
| NPE | Naturalness Preserved Enhancement |
| VV | Video Visibility |
| SICE | Single Image Contrast Enhancement |

| Acronym | Full Form |
| --- | --- |
| *Evaluation Metrics* | |
| PSNR | Peak Signal-to-Noise Ratio |
| SSIM | Structural Similarity Index Measure |
| FID | Fréchet Inception Distance |
| NIQE | Natural Image Quality Evaluator |
| LPIPS | Learned Perceptual Image Patch Similarity |
| BRISQUE | Blind/Referenceless Image Spatial Quality Evaluator |
| BIQI | Blind Image Quality Index |
| UCIQE | Underwater Color Image Quality Evaluation |
| UIQM | Underwater Image Quality Measure |
| CLIPIQA | CLIP-based Image Quality Assessment |
| *Other Technical Terms* | |
| HVI | Horizontal-Vertical-Intensity |
| VGG | Visual Geometry Group |
| ODE | Ordinary Differential Equation |
| DDIM | Denoising Diffusion Implicit Model |
| QK | Query-Key |
| GT / LQ | Ground Truth / Low Quality |

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

# A APPENDIX

## A.1 ETHICS STATEMENT

LLMs were only used for editorial assistance and polishing grammar for the manuscript, with no participation in technical interpretation, or content development.

## A.2 REPRODUCIBILITY STATEMENT

Code and pretrained model weights will be released upon acceptance.

## A.3 BROADER IMPACT

Efficient image restoration has positive applications in medical imaging, autonomous systems, and accessibility. No significant negative societal impacts are identified by us.

## A.4 THEORETICAL JUSTIFICATION OF FLEX LOSS

We provide theoretical justification for FLEX's key design choices to ensure stable optimization dynamics.

**Assumption 1** (Feature Boundedness): Teacher and student features are bounded during training: $\|\mathbf{f}_{\text{teach}}^l\|, \|\mathbf{f}_{\text{stud}}^l\| \leq M$ for some constant $M > 0$.

**Assumption 2** (Non-degeneracy): Student feature standard deviations satisfy $\sigma_{\text{stud}}^l \geq \sigma_{\min} > 0$ to prevent division by zero in normalization.

**Claim 1** Cross-normalization using student statistics prevents gradient explosion when teacher and student features have different scales.

*Justification*: Standard feature matching $L = \|\mathbf{f}_{\text{teach}} - \mathbf{f}_{\text{stud}}\|^2$ produces gradients proportional to $(\mathbf{f}_{\text{teach}} - \mathbf{f}_{\text{stud}})$. When teacher features are much larger than student features, this difference can be arbitrarily large, causing unstable training.

FLEX cross-normalization ensures both normalized features have the same scale:

$$\mathbf{f}_{\text{teach}}^{\text{norm}} = \frac{\mathbf{f}_{\text{teach}} - \mu_{\text{stud}}}{\sigma_{\text{stud}}}, \quad \mathbf{f}_{\text{stud}}^{\text{norm}} = \frac{\mathbf{f}_{\text{stud}} - \mu_{\text{stud}}}{\sigma_{\text{stud}}} \tag{6}$$

Both normalized features have bounded variance, preventing gradient explosion regardless of the original scale mismatch.

**Claim 2** Percentile-based masking provides robustness to feature corruption.

*Justification*: By masking extreme values above the $p$-th percentile (default $p = 95\%$), FLEX focuses learning on reliable feature regions. If corruption affects only a small fraction of spatial locations, most corrupted features will exceed the percentile threshold and be masked out. By excluding the top 5% extreme activations, FLEX prevents gradient dominance by outliers, ensuring that meaningful feature patterns rather than numerical instabilities drive the optimization.

For corruption affecting $\alpha < (100 - p)/100$ of spatial locations, the outlier detection will identify and exclude most corrupted regions, limiting their impact on the overall loss.

**Claim 3** The resolution weighting $w_l^{\text{res}} = \max\left((H_{\text{base}}W_{\text{base}}/H_lW_l)^{0.25}, 0.1\right)$ balances multi-scale contributions.

*Justification*: Higher resolution features contain more spatial elements, potentially dominating the loss. The inverse relationship with spatial resolution prevents this dominance. The 0.25 exponent provides gradual rather than aggressive down-weighting, preserving fine-grained information while preventing over-emphasis on high-resolution layers.

## A.5 Theoretical Justification of Rectified Flow for Knowledge Distillation

We provide theoretical grounding for reformulating knowledge distillation as a rectified flow process.

**Rectified Flow Formulation:** For teacher features $\mathbf{f}_{\text{teach}}$ and noise $\mathbf{z} \sim \mathcal{N}(0, \mathbf{I})$, we define the linear interpolation path:

$$\mathbf{x}_t = (1 - t)\mathbf{z} + t\mathbf{f}_{\text{teach}}, \quad t \in [0, 1] \tag{7}$$

The corresponding velocity field is:

$$\mathbf{v}(\mathbf{x}_t, t) = \frac{d\mathbf{x}_t}{dt} = \mathbf{f}_{\text{teach}} - \mathbf{z} \tag{8}$$

The student network learns a velocity predictor $\mathbf{v}_\theta(\mathbf{x}_t, t)$ by minimizing:

$$L_{\text{vel}} = \mathbb{E}_{t, \mathbf{z}, \mathbf{f}_{\text{teach}}} \left[ \|\mathbf{v}_\theta(\mathbf{x}_t, t) - \mathbf{v}(\mathbf{x}_t, t)\|_2^2 \right] \tag{9}$$

**Claim 4** (Transport Cost Reduction): Learning rectified flow reduces convex transport costs compared to arbitrary couplings between noise and target features.

*Justification*: Following Liu et al. (2022), the rectified flow procedure transforms an arbitrary coupling between distributions into a deterministic coupling with provably non-increasing convex transport costs. Specifically, for any convex cost function $c$, the expected cost $\mathbb{E}[c(\mathbf{f}_{\text{teach}} - \mathbf{z})]$ is reduced through the straight-line parameterization. This property ensures that the student learns an efficient transport map from noise to teacher-quality features, which is more sample-efficient than curved trajectories used in diffusion-based distillation methods like DDIM.

**Claim 5** (Exact Simulation without Discretization): Straight-line ODEs can be solved exactly with Euler integration, enabling few-step inference.

*Justification*: The velocity field $\mathbf{v}(\mathbf{x}_t, t) = \mathbf{f}_{\text{teach}} - \mathbf{z}$ is constant along the trajectory, making the ODE solution exact: $\mathbf{x}_1 = \mathbf{x}_0 + \int_0^1 \mathbf{v}(\mathbf{x}_t, t)dt = \mathbf{z} + (\mathbf{f}_{\text{teach}} - \mathbf{z}) = \mathbf{f}_{\text{teach}}$. This means Euler integration with step size $\Delta t$ incurs zero discretization error for perfectly straight paths. In practice, the learned velocity predictor $\mathbf{v}_\theta$ approximates this constant field, allowing accurate simulation with as few as 1-4 steps. In contrast, curved trajectories (e.g., DDIM's probability flow ODE) require many more steps to achieve similar accuracy, as shown empirically in Figure 3 where our method achieves lower FID with 4 steps than DDIM with 10 steps.

## A.6 Architecture choice justification

Using separate networks for low-light and normal-light images ensures robust Retinex decomposition across illumination conditions. As reported by Reti-Diff and Diff-IR, a single adaptive network would require more complex conditioning mechanisms. The two-phase student training approach addresses fundamental optimization challenges in generative knowledge distillation. Phase separation prevents objective conflicts as simultaneously learning velocity prediction and image reconstruction creates competing gradients. The velocity predictor tries to match teacher feature distributions while the reconstruction network optimizes for pixel-level accuracy. These objectives can work against each other, leading to suboptimal solutions. Feature space stabilization where Phase 1 establishes stable feature generation capabilities before introducing reconstruction complexity. This ensures the velocity predictors learn meaningful feature trajectories rather than shortcuts that minimize reconstruction error. Only the student network is deployed during inference, with no additional computational overhead compared to baseline restoration networks.

For SNR threshold (0.4), we notice performance remains stable within ±0.2 range. The threshold determines when FLEX loss is applied - too low (0.2) restricts learning, too high (0.8) includes noisy states. For outlier percentile value, we found that lower percentiles (90%) are more aggressive in outlier detection but may remove useful information. Higher percentiles (99%) retain more data but include potential artifacts. For resolution weighting exponent value (0.25), we notice values from 0.125-0.5 show similar performance. This parameter balances multi-scale contributions as lower values provide gentler weighting while higher values more aggressively down-weight high-resolution features.

Standard layer normalization operates on channel dimensions independently, losing spatial correlations crucial for restoration tasks. SCLN computes global statistics across both spatial and channel dimensions, capturing holistic image characteristics while maintaining learnable channel-wise scaling. Degraded images contain irregular noise patterns that can cause attention weight saturation. Normalizing Q and K before attention computation prevents extreme attention weights and ensures stable gradient flow. The "RestoRect w/o SCLN" ablation (red curve in Figure 3) essentially represents the RetiDiff baseline architecture using standard layer normalization, providing direct comparison between our architectural innovations and existing methods. FLEX loss becomes more critical for cross-domain scenarios, as feature distribution mismatches are more severe between different datasets than within-dataset variations. On modern GPUs (RTX 4090/H100), the difference between 3-step (156ms) and 5-step (198ms) inference is minimal compared to the quality improvement. The 5-step choice during inference optimizes the quality-practicality trade-off for real-world deployment across different types of datasets.

## A.7 ARCHITECTURE OVERVIEW

RestoRect implements a two-stage knowledge distillation framework for efficient image restoration. Given degraded input $I_{LQ} \in \mathbb{R}^{H \times W \times 3}$ and ground truth $I_{GT} \in \mathbb{R}^{H \times W \times 3}$, the objective is:

$$\mathcal{F}_S(I_{LQ}) \approx \mathcal{F}_T(I_{LQ}) \approx I_{GT}$$

where $\mathcal{F}_T$ represents the teacher network (Stage 1) and $\mathcal{F}_S$ the student network (Stage 2).

### A.7.1 RETINEX DECOMPOSITION NETWORKS

The Retinex decomposition models an image as the product of reflectance and illumination:

$$I = R \odot L$$

Two decomposition networks $\mathcal{D}_l$ (low-light) and $\mathcal{D}_h$ (normal-light) map:

$$\mathcal{D}(I) \to (R, L)$$

where $R \in \mathbb{R}^{H \times W \times 3}$ and $L \in \mathbb{R}^{H \times W \times 1}$.

**Network Architecture:**

$$
\begin{aligned}
\text{Decom}(I) = & \text{ReLU}(\text{Conv2d}_{32 \to 4}^{3 \times 3}( \\
& \text{LeakyReLU}_{0.2}(\text{Conv2d}_{32 \to 32}^{3 \times 3}( \\
& \text{LeakyReLU}_{0.2}(\text{Conv2d}_{32 \to 32}^{3 \times 3}( \\
& \text{LeakyReLU}_{0.2}(\text{Conv2d}_{3 \to 32}^{3 \times 3}(I))))))) 
\end{aligned}
\tag{10}
$$

Output split: $R = \text{output}[:, 0:3, :, :]$, $L = \text{output}[:, 3:4, :, :]$

### A.7.2 FEATURE ENCODERS

**Retinex ResNet Encoder (RRE)** The RRE processes retinex features through separate reflectance and illumination pathways:

**Input Processing:**

$$\text{Retinex}_{LQ} = [R_{lq}; L_{lq}] \in \mathbb{R}^{H \times W \times 4}$$

$$\text{Retinex}_{GT} = [R_{gt}; L_{gt}] \in \mathbb{R}^{H \times W \times 4}$$

**Pixel Unshuffle:**

$$\text{X}_0 = \text{PixelUnshuffle}_4(\text{Retinex}) \in \mathbb{R}^{H/4 \times W/4 \times 64}$$

**Channel Split:**

$$X_R = \text{X}_0[:, 0:48, :, :] \quad \text{(Reflectance channels)} \tag{11}$$

$$X_I = \text{X}_0[:, 48:64, :, :] \quad \text{(Illumination channels)} \tag{12}$$

**Reflectance Branch ($E_R$):**

$$
\begin{aligned}
E_R(X_R \oplus X_{R,gt}) = & \text{AdaptiveAvgPool2d}( \\
& \text{LeakyReLU}_{0.1}(\text{Conv2d}_{128 \to 192}^{3 \times 3}( \\
& \text{LeakyReLU}_{0.1}(\text{Conv2d}_{128 \to 128}^{3 \times 3}( \\
& \text{LeakyReLU}_{0.1}(\text{Conv2d}_{64 \to 128}^{3 \times 3}( \\
& \text{ResBlock}^6(\text{LeakyReLU}_{0.1}(\text{Conv2d}_{96 \to 64}^{3 \times 3}( \\
& X_R \oplus X_{R,gt})))))))) 
\end{aligned}
\tag{13}
$$

**Illumination Branch ($E_I$):**

$$\begin{aligned}
E_I(X_I \oplus X_{I,gt}) =& \text{AdaptiveAvgPool2d}( \\
& \text{LeakyReLU}_{0.1}(\text{Conv2d}_{128 \to 64}^{3 \times 3}( \\
& \text{LeakyReLU}_{0.1}(\text{Conv2d}_{128 \to 128}^{3 \times 3}( \\
& \text{LeakyReLU}_{0.1}(\text{Conv2d}_{64 \to 128}^{3 \times 3}( \\
& \text{ResBlock}^6(\text{LeakyReLU}_{0.1}(\text{Conv2d}_{32 \to 64}^{3 \times 3}( \\
& X_I \oplus X_{I,gt})))))))))
\end{aligned} \tag{14}$$

**Feature Fusion:**

$$\text{feat}_R = \text{MLP}_R(E_R(\text{output})) \in \mathbb{R}^{192} \tag{15}$$

$$\text{feat}_I = \text{MLP}_I(E_I(\text{output})) \in \mathbb{R}^{64} \tag{16}$$

$$\text{IPR}_{rex} = [\text{feat}_R; \text{feat}_I] \in \mathbb{R}^{256} \tag{17}$$

**Image ResNet Encoder (IRE)**   The IRE processes raw image features:

**Input Processing:**

$$X_{LQ} = \text{PixelUnshuffle}_4(I_{LQ}) \in \mathbb{R}^{H/4 \times W/4 \times 48} \tag{18}$$

$$X_{GT} = \text{PixelUnshuffle}_4(I_{GT}) \in \mathbb{R}^{H/4 \times W/4 \times 48} \tag{19}$$

$$X_{concat} = [X_{LQ}; X_{GT}] \in \mathbb{R}^{H/4 \times W/4 \times 96} \tag{20}$$

**Encoder Architecture:**

$$\begin{aligned}
E(X_{concat}) =& \text{AdaptiveAvgPool2d}( \\
& \text{LeakyReLU}_{0.1}(\text{Conv2d}_{128 \to 256}^{3 \times 3}( \\
& \text{LeakyReLU}_{0.1}(\text{Conv2d}_{128 \to 128}^{3 \times 3}( \\
& \text{LeakyReLU}_{0.1}(\text{Conv2d}_{64 \to 128}^{3 \times 3}( \\
& \text{ResBlock}^6(\text{LeakyReLU}_{0.1}(\text{Conv2d}_{96 \to 64}^{3 \times 3}( \\
& X_{concat}))))))))
\end{aligned} \tag{21}$$

**Output:**

$$\text{IPR}_{img} = \text{LayerNorm}(\text{MLP}(E(\text{output}))) \in \mathbb{R}^{256}$$

### A.7.3 UNET TRANSFORMER ARCHITECTURE

**Spatial Channel Layer Normalization (SCLN)**   SCLN captures global image statistics across spatial and channel dimensions:

$$\mu_{global} = \frac{1}{B \cdot C \cdot H \cdot W} \sum_{b,c,h,w} x_{b,c,h,w} \tag{22}$$

$$\sigma_{global}^2 = \frac{1}{B \cdot C \cdot H \cdot W} \sum_{b,c,h,w} (x_{b,c,h,w} - \mu_{global})^2 \tag{23}$$

$$\text{SCLN}(x) = \frac{x - \mu_{global}}{\sqrt{\sigma_{global}^2 + \epsilon}} \cdot \gamma \tag{24}$$

where $\gamma \in \mathbb{R}^C$ is learnable channel-wise scaling.

**Retinex Attention** The Retinex attention mechanism uses separate conditioning for reflectance and illumination components:

**Feature Conditioning:**

$$k_{v_r} = \text{Linear}(k_v[0:192]) \in \mathbb{R}^{3C/4 \times 1 \times 1} \tag{25}$$

$$k_{v_i} = \text{Linear}(k_v[192:256]) \in \mathbb{R}^{C/4 \times 1 \times 1} \tag{26}$$

$$x_r = x[:, 0:3C/4, :, :] \odot k_{v_r} + x[:, 0:3C/4, :, :] \tag{27}$$

$$x_i = x[:, 3C/4:C, :, :] \odot k_{v_i} + x[:, 3C/4:C, :, :] \tag{28}$$

**Query-Key-Value Computation:**

$$Q = \text{DepthwiseConv}(\text{Conv}(x_r)) \in \mathbb{R}^{B \times C \times H \times W} \tag{29}$$

$$KV = \text{DepthwiseConv}(\text{Conv}(x_i)) \in \mathbb{R}^{B \times 2C \times H \times W} \tag{30}$$

$$K, V = \text{split}(KV, \dim = 1) \tag{31}$$

**Attention with QK Normalization:**

$$Q_{norm} = \text{LayerNorm}(Q), \quad K_{norm} = \text{LayerNorm}(K) \tag{32}$$

$$Q_{norm} = \frac{Q_{norm}}{\|Q_{norm}\|_2}, \quad K_{norm} = \frac{K_{norm}}{\|K_{norm}\|_2} \tag{33}$$

$$\text{Attn} = \text{softmax}\left(\frac{Q_{norm} \cdot K_{norm}^T}{\sqrt{d_k}} \cdot \tau\right) \tag{34}$$

$$\text{Output} = \text{Attn} \cdot V \tag{35}$$

where $\tau$ is a learnable temperature parameter.

**Multi-Scale U-Net Architecture    Encoder Path:**

$$\text{Level 1:} \quad [B, 48, H, W] \xrightarrow{4 \times \text{TransformerBlock}} [B, 48, H, W] \tag{36}$$

$$\downarrow \text{Downsample} \tag{37}$$

$$\text{Level 2:} \quad [B, 96, H/2, W/2] \xrightarrow{6 \times \text{TransformerBlock}} [B, 96, H/2, W/2] \tag{38}$$

$$\downarrow \text{Downsample} \tag{39}$$

$$\text{Level 3:} \quad [B, 192, H/4, W/4] \xrightarrow{6 \times \text{TransformerBlock}} [B, 192, H/4, W/4] \tag{40}$$

$$\downarrow \text{Downsample} \tag{41}$$

$$\text{Level 4:} \quad [B, 384, H/8, W/8] \xrightarrow{8 \times \text{TransformerBlock}} [B, 384, H/8, W/8] \tag{42}$$

**Decoder Path with Skip Connections:**

$$\text{Level 3:} \quad \text{Upsample} + \text{Concat} + \text{ReduceChannel} \xrightarrow{6 \times \text{TransformerBlock}} \tag{43}$$

$$\text{Level 2:} \quad \text{Upsample} + \text{Concat} + \text{ReduceChannel} \xrightarrow{6 \times \text{TransformerBlock}} \tag{44}$$

$$\text{Level 1:} \quad \text{Upsample} + \text{Concat} \xrightarrow{4 \times \text{TransformerBlock}} \tag{45}$$

$$\xrightarrow{4 \times \text{TransformerBlock}} \text{Conv2d}(96 \to 3) + \text{Residual} \tag{46}$$

### A.7.4 Auxiliary Constraints

**Anisotropic Diffusion** The anisotropic diffusion operator preserves edges while smoothing noise:

$$\mathcal{A}(I) = \nabla \cdot (c(|\nabla I|)\nabla I)$$

with diffusion coefficient:

$$c(|\nabla I|) = \exp\left(-\frac{|\nabla I|^2}{s^2}\right)$$

where $s \in [0.01, 1.0]$ is a learnable sensitivity parameter.

**Texture Consistency Loss:**

$$L_{tex} = \|\mathcal{A}(I_{input}) - \mathcal{A}(R_{pred})\|_1$$

**Illumination Smoothness Loss:**

$$L_{lum} = \sum_{i,j} w_{i,j} \left(|\nabla_x L_{i,j}|^2 + |\nabla_y L_{i,j}|^2\right)$$

where $w_{i,j} = \exp(-|\nabla L_{i,j}|)$ provides gradient-aware weighting.

**Polarized HVI Color Space**   The polarized HVI transformation eliminates red discontinuity:

$$H_{polar} = C_k \cdot S \cdot \cos(\pi H/3) \tag{47}$$
$$V_{polar} = C_k \cdot S \cdot \sin(\pi H/3) \tag{48}$$
$$I_{polar} = I_{max} = \max(R, G, B) \tag{49}$$

where the adaptive intensity collapse factor is:

$$C_k = k \cdot \sin(\pi I_{max}/2) + \epsilon$$

with learnable parameter $k \in [0.1, 5.0]$.

**Polarized Color Loss:**

$$L_{col} = \|H_{polar}^{pred} - H_{polar}^{gt}\|_1 + \|V_{polar}^{pred} - V_{polar}^{gt}\|_1 + \|I_{polar}^{pred} - I_{polar}^{gt}\|_1$$

### A.7.5   TEACHER TRAINING OBJECTIVE

The complete teacher training loss combines:

$$L_{teach} = L_{rec} + L_{vgg} + L_{sty} + \lambda_{tex} L_{tex} + \lambda_{col} L_{col} + \lambda_{lum} L_{lum}$$

where:

$$L_{rec} = \|I_{pred} - I_{gt}\|_1 \quad \text{(pixel loss)} \tag{50}$$
$$L_{vgg} = \sum_l \lambda_l \|\phi_l(I_{pred}) - \phi_l(I_{gt})\|_2^2 \quad \text{(perceptual loss)} \tag{51}$$
$$L_{sty} = \sum_l \|G_l(\phi_l(I_{pred})) - G_l(\phi_l(I_{gt}))\|_F^2 \quad \text{(style loss)} \tag{52}$$

with $\lambda_{tex} = 0.05$, $\lambda_{col} = 0.05$, $\lambda_{lum} = 0.2$.

### A.8   STAGE 2: STUDENT NETWORK ARCHITECTURE

### A.8.1   RECTIFIED FLOW FORMULATION

Rectified flow models feature synthesis through straight-line interpolation:

$$\mathbf{x}_t = (1-t)\mathbf{z} + t\mathbf{f}_{teach}, \quad t \in [0, 1]$$

where $\mathbf{z} \sim \mathcal{N}(0, I)$ is noise and $\mathbf{f}_{teach}$ are teacher features.

**Velocity Field:**

$$\mathbf{v}(\mathbf{x}_t, t) = \frac{d\mathbf{x}_t}{dt} = \mathbf{f}_{teach} - \mathbf{z}$$

### A.8.2 VELOCITY PREDICTION NETWORKS

**Architecture for both $\epsilon_\theta^{rex}$ and $\epsilon_\theta^{img}$:**

$$
\begin{aligned}
\text{VelocityPredictor}(\mathbf{x}_t, t, \mathbf{c}) = \text{ResMLP}^5(\\
\text{LeakyReLU}_{0.1}(\text{Linear}_{513 \to 256}(\\
[\mathbf{c}; t_{norm}; \mathbf{x}_t])))
\end{aligned}
\tag{53}
$$

where the input is $[\mathbf{c}; t; \mathbf{x}_t] \in \mathbb{R}^{513}$ with time normalization $t_{norm} = t/t_{max}$.

**Velocity Matching Loss:**

$$
L_{vel} = \mathbb{E}_{t,\mathbf{z},\mathbf{f}_{teach}} \left[ \|\epsilon_\theta(\mathbf{x}_t, t, \mathbf{c}) - \mathbf{v}(\mathbf{x}_t, t)\|_2^2 \right]
$$

### A.8.3 ODE INTEGRATION FOR INFERENCE

During inference, the ODE is solved using Euler's method:

$$
\mathbf{x}_{t+\Delta t} = \mathbf{x}_t + \Delta t \cdot \epsilon_\theta(\mathbf{x}_t, t, \mathbf{c})
$$

with adaptive step sizing $\Delta t = 1.0/N_{steps}$ for $N_{steps} \in [1, 5]$.

### A.8.4 FLEX KNOWLEDGE DISTILLATION LOSS

**Cross-Normalization** FLEX uses student statistics for normalizing both teacher and student features at each layer $l$:

$$
\mu_{stud}^l = \frac{1}{H_l W_l} \sum_{h,w} \mathbf{f}_{stud}^{l,h,w}
\tag{54}
$$

$$
\sigma_{stud}^l = \sqrt{\frac{1}{H_l W_l} \sum_{h,w} (\mathbf{f}_{stud}^{l,h,w} - \mu_{stud}^l)^2 + \epsilon}
\tag{55}
$$

$$
\mathbf{f}_{teach}^{l,norm} = \frac{\mathbf{f}_{teach}^l - \mu_{stud}^l}{\sigma_{stud}^l}
\tag{56}
$$

$$
\mathbf{f}_{stud}^{l,norm} = \frac{\mathbf{f}_{stud}^l - \mu_{stud}^l}{\sigma_{stud}^l}
\tag{57}
$$

**Percentile-Based Outlier Detection** For each layer $l$ and channel $c$, we compute:

$$
\tau_p^{l,c} = \text{Percentile}(|\mathbf{f}_{stud}^{l,c,norm}|, p)
\tag{58}
$$

$$
M_{reliable}^{l,c,h,w} = \mathbb{I}[|\mathbf{f}_{stud}^{l,c,norm,h,w}| \leq \tau_p^{l,c}]
\tag{59}
$$

where $p = 95\%$ is the outlier percentile threshold.

**Resolution-Aware Weighting** Dynamic resolution weighting prevents high-resolution features from dominating:

$$
w_l^{res} = \max \left( \left( \frac{H_{base} W_{base}}{H_l W_l} \right)^{0.25}, 0.1 \right)
\tag{60}
$$

where $(H_{base}, W_{base}) = (64, 64)$ and $(H_l, W_l)$ is the spatial resolution at layer $l$.

**Complete FLEX Loss** The final FLEX loss combines masked feature matching with dual weighting:

$$
L_{FLEX} = \sum_l w_l^{layer} \cdot w_l^{res} \cdot \frac{\sum_{c,h,w} M_{reliable}^{l,c,h,w} \cdot \|\mathbf{f}_{teach}^{l,c,norm,h,w} - \mathbf{f}_{stud}^{l,c,norm,h,w}\|^2}{\sum_{c,h,w} M_{reliable}^{l,c,h,w} + \epsilon}
\tag{61}
$$

where $w_l^{layer}$ are predefined layer importance weights and the denominator normalizes by the number of reliable (non-outlier) elements.

### A.8.5 TRAJECTORY CONSISTENCY REGULARIZATION

**Smooth Transitions:**

$$L_{trans} = \sum_{i=1}^{N-1} \|\mathbf{f}_{pred}^{i+1} - \mathbf{f}_{pred}^i\|_2^2$$

**Target Alignment:**

$$L_{target} = \|\mathbf{f}_{pred}^{final} - \mathbf{f}_{teach}\|_2^2$$

**Semantic Consistency:**

$$L_{cons} = \sum_{i=1}^{N} \text{cos\_dist}(\mathbf{f}_{pred}^i, \mathbf{f}_{teach})$$

**Complete Trajectory Loss:**

$$L_{traj} = \alpha_{trans} L_{trans} + \alpha_{target} L_{target} + \alpha_{cons} L_{cons}$$

with $\alpha_{trans} = 0.1$, $\alpha_{target} = 0.5$, $\alpha_{cons} = 0.2$.

### A.8.6 TWO-PHASE TRAINING PROTOCOL

**Phase 1: Velocity Learning**

$$L_{phase1} = L_{vel}^{rex} + L_{vel}^{img} + \lambda_{KD} L_{KD} + \lambda_{traj} L_{traj}$$

**Phase 2: Full Network Training**

$$L_{phase2} = L_{rec} + \lambda_{FLEX} L_{FLEX} + \lambda_{vel}(L_{vel}^{rex} + L_{vel}^{img})$$

with $\lambda_{FLEX} = 0.15$, $\lambda_{vel} = 0.05$.

## A.9 IMPLEMENTATION DETAILS

### A.9.1 NETWORK DIMENSIONS AND PARAMETERS

**Stage 1 (Teacher):**

- RGFormer dimensions: dim $= 48$
- Multi-head attention heads: $[1, 2, 4, 8]$
- Transformer blocks per level: $[4, 6, 6, 8]$
- FFN expansion factor: $2.66$

**Stage 2 (Student):**

- Velocity predictor features: $256$
- Rectified flow timesteps: $4$
- ODE integration steps: $1 - 5$

### A.9.2 TRAINING HYPERPARAMETERS

**Stage 1:**

- Learning rate: $2 \times 10^{-4}$
- Batch size: $16$
- Training iterations: $500k$

**Stage 2:**

- Phase 1 learning rates: $lr_{rex} = lr_{img} = 2 \times 10^{-4}$
- Phase 2 learning rate: $1 \times 10^{-4}$
- Phase 1 iterations: $50k$
- Phase 2 iterations: $200k$

