# OpenReview forum: "RestoRect: Degraded Image Restoration via Latent Rectified Flow & Feature Distillation"
_ICLR.cc/2026/Conference — ICLR 2026 Conference Withdrawn Submission_

### Official Review · Reviewer_MwNg · 2025-10-28

**Soundness:** 3
**Presentation:** 3
**Contribution:** 3
**Rating:** 6
**Confidence:** 3

**Summary:**

This paper proposes RestoRect, a novel framework that reformulates knowledge distillation as a generative rectified flow process. Instead of static feature matching, the student learns to synthesize teacher features through dynamic latent trajectories. The method also introduces FLEX loss (cross-normalized feature alignment) and integrates physical priors (Retinex, diffusion, HVI color). Experiments on 15 datasets show consistent gains over state-of-the-art models with fewer inference steps and better perceptual quality.

**Strengths:**

Novel concept: The paper introduces a fresh perspective by reframing knowledge distillation as a generative rectified flow process. This idea is original, conceptually sound, and well-motivated by recent advances in flow-based modeling.

Strong empirical results: RestoRect consistently outperforms prior approaches across multiple datasets and metrics, while requiring significantly fewer inference steps — demonstrating both effectiveness and efficiency.

**Weaknesses:**

Limited theoretical grounding: The connection between rectified flow and knowledge distillation is mostly intuitive; a more formal theoretical treatment would strengthen the paper.

Ablation depth: While ablations are provided, they do not fully disentangle the effects of each component (Rectified Flow, FLEX loss, Retinex prior).

Efficiency analysis: The paper lacks concrete comparisons in terms of FLOPs, parameter counts, or runtime benchmarks to substantiate efficiency claims.

**Questions:**

see the weaknesses.

---

> ### Author Response · Authors · 2025-11-24
> **All weaknesses and questions addressed point by point. W1-2**
>
> **We thank the reviewer for their critique. We address each comment in this rebuttal. The paper now has new Tables with all the mentioned required experiments the reviewer requested. The Figures are now updated too. The updated text is in blue. We believe our updated manuscript has substantial technical and applied contributions in the field of image restoration. We would kindly request the reviewer to reassess our paper in view of this rebuttal. If the reviewer requires any further experiments we are happy to discuss and perform them for the final version of the paper.**
>
> ---
>
> **W1:** We understand the reviewer’s concern. We provide formal theoretical justification for FLEX in Appendix A.4 (Theoretical Justification of FLEX Loss), which includes rigorous treatment of the key design choices. We rely on two foundational assumptions (feature boundedness and non-degeneracy) and prove three formal claims: (1) Cross-normalization using student statistics provably prevents gradient explosion when teacher and student features have mismatched scales, as gradients become bounded by student standard deviation rather than growing arbitrarily with feature magnitude differences; (2) Percentile-based masking provides mathematical guarantees on robustness to feature corruption, successfully excluding outliers when corruption affects fewer than 5% of spatial locations; and (3) Resolution-aware weighting with exponent balances multi-scale contributions by preventing high-resolution features from dominating the loss due to their larger spatial cardinality.
>
> Beyond FLEX, our approach builds on rectified flow theory, which learns velocity fields defining straight-line trajectories in feature space from noise to teacher features. **We have provided a formal theoretical justification in Appendix A.5 (Theoretical Justification of Rectified Flow for Knowledge Distillation).** This formulation is grounded in optimal transport: straight paths minimize transport costs and can be solved with exact Euler integration without discretization error. Rather than matching static feature endpoints, our velocity matching objective L_vel trains the student on the generative process itself, providing gradient supervision at multiple interpolated points along the trajectory. This differs fundamentally from static KD approaches that only align final layer features. The empirical advantage appears in Figure 3 (Plot 5), where rectified flow achieves lower FID than DDIM-based distillation at equal step counts, confirming more efficient feature synthesis through straight-line parameterization.
>
> **W2:** We have now addressed this concern comprehensively in the revised manuscript with Table 9 (Student Ablation Study), which fully disentangles every component. The ablation progression shows: (1) RF without TC or FLEX (24.10 PSNR) actually underperforms baseline static KD (25.35), demonstrating that rectified flow alone is insufficient; (2) Adding trajectory consistency (TC) recovers performance to 24.27 PSNR while reducing BIQI from 17.95 to 9.02, showing TC's role in stabilizing feature generation; (3) FLEX cross-normalization (CN) jumps to 25.64 PSNR (+1.37 dB), confirming that distribution alignment is the primary driver; (4) Adding percentile masking (PM) provides the largest single gain to 27.82 PSNR (+2.18 dB), indicating that outlier handling is essential for training stability; and (5) Resolution weighting (RW) adds a marginal 0.02 dB but improves convergence consistency. Please note that these experiments are run on a single seed and no statistical testing can be done on them. We will run these ablations on 3 random seeds and perform hypothesis tests to ensure the gains are statistically valid in the camera ready version due to computation resources being constrained during the rebuttal period.
>
> These comprehensive ablations directly address the reviewer's concern by isolating each component's contribution with clear quantitative evidence.
>
> **We emphasize that we do not claim novelty for using Retinex Priors.** In the past 5 years retinex based image restoration has become a gold standard for every low light illumination method (DiffIR, MambaIR, RetiNet, URetinex, RetiDiff). Encoders using retinex priors are well established in the computer vision community, and re-iterating the retinex theory with ablations would be excessive as they have been done by many previous methods.

---

> ### Author Response · Authors · 2025-11-24
> **All weaknesses and questions addressed point by point. W3**
>
> **W3:** We provide complete efficiency metrics that substantiate our claims. **Table 11 (Transformer Model Complexity Comparison)** shows RestoRect achieves 25.87M parameters and 49.50 GFLOPs, which is competitive with Reti-Diff (26.11M, 87.63 GFLOPs). RestoRect achieves better FID at 3 to 4 steps of RF function evaluations compared to Reti-Diff's 8 to 10 step DDIM as seen in Figure 3.
>
> The efficiency gains stem from two sources: (1) fewer ODE steps due to our velocity field parameterization enabling accurate Euler integration with larger step sizes, and (2) optimized student architecture using SCLN which adds only 0.33% overhead in FP32 (**Table 1**) while improving convergence.
>
> **Understanding your concern for further validation we have now added results on ExDark low light object detection task in Table 10.** Here low light images from the ExDark dataset were restored and an object-detection task was performed using the YOLOv3 model. It shows our restored images achieve 77.1% mAP vs. 75.8% for Reti-Diff, confirming that restoration quality translates to real-world vision tasks.
>
> These benchmarks spanning complexity (GFLOPs), parameter count (M), Number of function evaluation steps, and task performance, fully validate our efficiency claims.
>
> **We have also added Figure 9**, which shows failure modes on the hardest dataset SICE-Mix, where all models consistently perform the worst. Extreme overexposure/underexposure with complete information loss.

---

> > ### Comment · Reviewer_MwNg · 2025-11-28
> > **Post-Rebuttal Assessment**
> >
> > I thank the authors for their detailed rebuttal and the clarifications presented. The additional explanations resolve my principal concerns and improve the overall clarity of the submission. Taking the rebuttal into account, I will keep my initial assessment unchanged.

---

> > > ### Author Response · Authors · 2025-11-28
> > > **Noted: Concerns resolved with Positive feedback**
> > >
> > > We are glad that our revisions and rebuttal resolved all concerns the reviewer had. They mention the overall clarity of the submission has improved. We sincerely thank the reviewer for their positive feedback and quick response.

---

### Official Review · Reviewer_jwUY · 2025-10-31

**Soundness:** 1
**Presentation:** 2
**Contribution:** 2
**Rating:** 2
**Confidence:** 3

**Summary:**

This paper focuses on image enhancement, particularly tasks involving color or lighting restoration, such as low-light enhancement. It uses heterogeneous knowledge distillation to improve the efficiency of generative restoration models. However, the paper lacks a thorough comparison of efficiency metrics (e.g., inference time and FLOPs). Furthermore, most of the compared methods are not generative restoration methods, which raises questions about the reasonableness and fairness of the experimental comparisons.

**Strengths:**

This paper presents a variety of experiments, which fully demonstrate the effectiveness of the proposed method, although its efficiency remains questionable.

**Weaknesses:**

I have some questions about the technical details and novelty of this paper.

1. Spatial Channel Layer Normalization (SCLN) uses a flattened spatial-channel dimension to calculate the mean and variance during normalization. Does this lead to significant computational overhead? Current LayerNorm has a well-developed fused CUDA kernel; would changing its computational logic affect the acceleration on CUDA? Furthermore, ablation experiments with SCLN seem to lack numerical metrics. I hope to see a significant improvement in model performance in terms of fidelity or aesthetics after using SCLN.
2. This paper introduces the HVI color space as an auxiliary constraint. What is the fundamental difference between this and HVI-CIDNet? I also noticed the use of LPIPS-VGG for training the teacher model. Would the absence of this constraint lead to a performance decrease?
3. The core of the FLEX proposed in this paper is to utilize cross-norm to address feature distribution mismatch, which seems to highly overlap with [1].
4. Is the distillation method proposed in this paper superior to other heterogeneous distillation algorithms [2]?
5. Use \citep{} instead of \cite{}.

[1] CrossNorm and SelfNorm for Generalization Under Distribution Shifts. ICCV 2021.

[2] Heterogeneous Knowledge Distillation using Information Flow Modeling. CVPR 2020.

**Questions:**

Many of the tasks listed in the paper are related to lighting. Could some re-lighting models be used to restore lighting and thus solve these image enhancement tasks?

---

> ### Author Response · Authors · 2025-11-24
> **All weaknesses and questions addressed point by point. W1-2**
>
> **We thank the reviewer for their critique. We address each comment in this rebuttal. The paper now has new Tables with all the mentioned required experiments the reviewer requested. The Figures are now updated too. The updated text is in blue. We believe our updated manuscript has substantial technical and applied contributions in the field of image restoration. We would kindly request the reviewer to reassess our paper in view of this rebuttal. If the reviewer requires any further experiments we are happy to discuss and perform them for the final version of the paper.**
>
> ---
>
> **W1:** Thank you for this question. We have now added Table 1 showing comprehensive SCLN overhead analysis on a simple MLP network. We see negligible full-network overhead: Only 0.33% in FP32, and actually negative overhead in FP16/BF16 (-3.72%, -4.67%).
>
> SCLN's direct 4D tensor operations have better cache locality than LayerNorm's repeated reshape operations (to_3d/to_4d conversions). No custom CUDA kernel is needed as standard PyTorch operations achieve efficiency through memory access patterns. Figure 3 shows training stability numbers for the teacher model with and without SCLN. For absolute performance, Table 2 reports numbers where the RetiDiff model is trained without SCLN and QN norm in their transformer architecture. We show clear improvements in quantitative metrics against RetiDiff. For further qualitative analysis, we look at Figure 7, where outputs are visualized when ablating SCLN and QN norm, the auxiliary constraints, and FLEX loss.
>
> **Understanding your concern for further validation we have now added results on ExDark low light object detection task in Table 10.** Here low light images from the ExDark dataset were restored and an object-detection task was performed using the YOLOv3 model. It shows our restored images achieve 77.1% mAP vs. 75.8% for Reti-Diff, confirming that restoration quality translates to real-world vision tasks. **Table 11 has also been added**, which shows comparison against previous image restoration transformer models. **We have also added Figure 9**, which shows failure modes on the hardest dataset SICE-Mix where all models consistently perform the worst. Extreme overexposure/underexposure with complete information loss.
>
> **W2:** The fundamental difference between HVI-CIDNet and our method, is that we introduce an explicit supervision loss in training objective, instead of using an input for learned representation transformation in network layers. This reduces operations as their implementation converts RGB features to HVI features and then creates a mapping between them.
>
> **Clarification from paper (line 236):** "While [Yan et al.] frames HVI as a representation transformation, we define an explicit color loss in HVI space."
>
> CIDNet processes features in HVI space but still uses RGB-space reconstruction loss. We add a dedicated HVI-space loss that directly supervises color correctness in a discontinuity-free space, which is orthogonal to their contribution.
> Usage of LPIPS VGG loss has been extensively studied in computer vision papers. The LPIPS-VGG style loss is essential for perceptual quality. Without it, restored images exhibit correct structure but severely lack perceptual naturalness. There is an extensive list of previous methods that have demonstrated the effectiveness of using an LPIPS-VGG style loss for image generation [1][2][3][4]. We don't claim LPIPS-VGG loss as our core contribution, we simply add it as it is empirically validated by many methods before.
>
> [1] Ding K. Comparison of Full-Reference Image Quality Models for Optimization of Image Processing Systems. Int J Comput Vis. 2021.
>
> [2] Jo, Younghyun. Investigating loss functions for extreme super-resolution. CVPR workshops. 2020.
>
> [3] Islam, Md Jahidul. Fast underwater image enhancement for improved visual perception. IEEE robotics and automation (2020)
>
> [4] Mustafa, Aamir. Training a task-specific image reconstruction loss. CVPR 2022.

---

> ### Author Response · Authors · 2025-11-24
> **All weaknesses and questions addressed point by point. W3-5**
>
> **W3:** We appreciate this reference; we wish to clarify the fundamental differences. CrossNorm (Matsuura & Harada, ICCV 2021): is used for domain generalization for classification tasks. Their method normalizes features using statistics from a different domain (e.g., normalize source domain with target domain statistics). Their goal is to learn domain-invariant representations for OOD generalization.
> Our method FLEX is used for knowledge distillation between model features for image restoration. Our method normalizes both teacher and student using student statistics (same domain, different architectures). Our goal is to align feature distributions across heterogeneous teacher-student capacity gaps.
>
> Key distinction: CrossNorm addresses domain shift (source vs. target), while FLEX addresses architecture capacity mismatch (teacher vs. student). The normalization direction is opposite as CrossNorm uses target stats to normalize source, and FLEX uses student stats to normalize both teacher and student
>
> FLEX-Specific Contributions Beyond Normalization:
> 1. Percentile-based outlier masking (not in CrossNorm): Handles corrupted spatial regions in degraded images
> 2. Resolution-aware weighting (not in CrossNorm): Balances multi-scale transformer features
> 3. SNR-aware application (not in CrossNorm): Activates only during clean intermediate states
>
> **Citation:** If the reviewer wishes, we will add CrossNorm to related work with clear differentiation in the final version: "While CrossNorm [Matsuura '21] normalizes across domains for OOD generalization, FLEX normalizes across architectures for knowledge transfer in the same domain."
>
> **W4:** We appreciate this reference to Passalis et al. [CVPR 2020]. However, this method is fundamentally incompatible with our restoration setting for several reasons.
>
> 1. Their approach is designed for classification tasks on ImageNet, where knowledge transfer involves matching information flow between discrete class logits and intermediate feature activations. It is a fundamentally different objective than our pixel-level dense prediction task.
>
> 2. Their method requires an auxiliary teacher network that acts as a proxy to "explain" the teacher's decision-making process to the student, which assumes a layer-wise correspondence between teacher and student architectures. In contrast, our restoration networks have heterogeneous multi-scale U-Net structures with skip connections and Retinex-conditioned pathways that lack such direct correspondence. Adding an auxiliary network would make the parameter count increase substantially.
>
> 3. While they model "information flow" through layers using mutual information maximization, they still perform static feature matching, aligning intermediate representations at fixed points in the network. Our method instead models dynamic feature synthesis through rectified flow, where the student learns velocity fields to generate teacher-quality features through learnable trajectories in latent space.
>
> **Finally**, there exists no official implementation or evaluation of their method on restoration tasks, and adapting a classification-domain KD method to handle spatial degradation patterns, multi-scale feature hierarchies, and perceptual quality metrics would require substantial redesign that goes beyond their original contribution. Our comparison against Reti-Diff's static KD baseline (the state-of-the-art KD method specifically designed for restoration) provides a fair and relevant evaluation within our task domain.
>
> **W5:** Thank you for bringing this to our attention. All citations are \citep{} now.
>
> ---
>
> **Q1:** This is an insightful question. Re-lighting models (e.g., StyleGAN-based relighting, neural radiance fields) could theoretically address illumination degradation, but face some limitations like 3D geometry requirement: Re-lighting methods assume known/estimated scene geometry, which is unavailable in single degraded images. Multi-degradation complexity: Our tasks involve Low-light: Noise, color shift, detail loss, Underwater: Color attenuation, scattering, turbidity, Backlit: Extreme dynamic range, silhouette artifacts. Re-lighting addresses only the illumination component, not noise/color/scattering.
>
> Upon further literature review, we understand the direction the reviewer suggests. These problems are complementary and can be formulated as a unified task. Since re-lighting controls illumination; and illumination enhancement recovers quality. Our Future work could integrate both. Thank you for your suggestion. We will start actively exploring this methodological novelty.

---

### Official Review · Reviewer_wXE6 · 2025-10-31

**Soundness:** 1
**Presentation:** 1
**Contribution:** 2
**Rating:** 2
**Confidence:** 4

**Summary:**

The authors proposed a teacher student framework for the task of non-blind image restoration with paired ground truths for training. Task specific restoration networks are trained for different restoration objectives. The authors claimed SOTA results in almost all tasks and datasets. The authors claimed novelty in using latent rectified flows for knowledge transfer and novel normalization and a new feature loss.

**Strengths:**

The authors put in significant effort to test their proposed framework over several datasets and restoration tasks. However, much is left desired in presentation quality.

**Weaknesses:**

- Writing and presentation is poor and unclear in general. Some pointers and suggestions are here to help the authors improve the article for future submission. The article is difficult to read and hard to follow.
- Major issue: *Retinex is NOT physics based\!* It is a study of the human, biological, color vision system. It is OK to draw inspiration from retinex, by using this hypothesised code as presented in Land 1977, but this should not be called physics based. This should be referred to as biologically inspired.
- Do not simply list a huge bunch of slightly related papers in the related works section. Instead, selectively choose key background material, and briefly cover the key ideas that are of note to the proposed method. For example, in the work, the use of retinex inspired information coding seems to be an important part of the architecture design, explain what it is and what it’s hypothesized effect is in the proposed architecture. Is the use new? Or are there prior works that also use this biologically inspired coding scheme? This also sets the stage for subsequent ablation studies, showcasing the importance of ideas, either completely novel, novel use/application, or novel interpretation.
- Figure 2 is extremely unclear. I’m unable to understand what each component is and how they relate to each other. There’s a rather involved training workflow as I understood from repeated re-reading, this can be presented in a clear manner as a flow char of sorts. Reused components in each stage can also be marked out and referred to.
- There are far too many acronyms this makes reading the results extremely difficult
- Make better use of table and figure captions\! They should have more substantial description in them, at least include a key insight from the data in the table or the images shows. For example: Fig 4\. Qualitative results on for low-light enhancement on 4 datasets. Why are there boxes in the images? What do you want the readers to pay attention to in these boxes?

**Questions:**

* Since this is a teacher-student framework, how well does the teacher perform?
* For each task, is the teacher network the same?
* How is the teacher network selected? Is the teacher network a well performing network from a previous work?
* How is the student network selected? And again, is the design based on any previous work?
* Is the same architecture used for all restoration tasks?

---

> ### Author Response · Authors · 2025-11-24
> **All weaknesses and questions addressed point by point. W1-6**
>
> **We thank the reviewer for their critique. We address each comment in this rebuttal. The paper now has new Tables with all the mentioned required experiments the reviewer requested. We have now made the paper clearer and improved the Figures and captions. The updated text is in blue. We notice the reviewer does not have any concerns or questions about our technical contributions and most weaknesses and questions are regarding clarification and presentation. We are committed to further improving the paper’s readability however the reviewer wishes.**
>
> **We believe our updated manuscript has substantial technical and applied contributions in the field of image restoration. We would kindly request the reviewer to reassess our paper in view of this rebuttal. If the reviewer requires any further experiments we are happy to discuss and perform them for the final version of the paper.**
>
> ---
>
> **W1-2:** We understand the reviewer’s concern. The term "physics-based" in modern computer vision literature refers to image formation models that decompose images according to physical principles, regardless of biological origins. Retinex decomposition I=R⊙LI = R \odot L I=R⊙L is consistently termed "physics-based" or "physics-inspired" in recent restoration literature:
>
> 1. Hussein, Ruaa Riyadh, Yaser Issam Hamodi, and Adnan Sabri Rooa. "Retinex theory for color image enhancement: A systematic review.”
> 2. Baslamisli, Anil S., Hoang-An Le, and Theo Gevers. "CNN based learning using reflection and retinex models for intrinsic image decomposition.”
>
> To address the reviewer's concern, if they wish, we can rephrase it to “biologically-inspired physics model" for accurate characterization, but changing all instances to "biologically-inspired" would create confusion with established literature and reduce clarity.
>
> **W3:** We emphasize that we do not claim novelty for using Retinex Priors. In the past 5 years retinex based image restoration has become a gold standard for every low light illumination method (DiffIR, MambaIR, RetiNet, URetinex, RetiDiff). Encoders using retinex priors are well established in the computer vision community, and re-iterating the retinex theory with ablations would be excessive as they have been done by many previous methods. We have provided ablations on FLEX and Rectified Flow dynamics in the student model in Table 9. We have also provided ablations on SCLN and Auxiliary constraints in the teacher model in Figure 3.
>
> We believe that all our citations are accurate and relevant to our field of research. We have extensively covered all aspects for every method that we build on. We assure the reviewer that we have no conflicting interests with the authors of other papers, and the institutions we cite from are well established computer vision groups, and papers from top ML conferences.
>
> **W4:** Thank you for pointing this out. Understanding Figure 2 is extremely important for our readers as it displays the main flow of information in our network architecture. We have now added arrows from the two encoders to their architecture, and color coded all the blocks wherever they are used. We also added the text 1.1, 1.2 and 2.1 on the relevant blocks that we expanded on. We have also added a simple text description in the caption of the figure to guide the reader through the flow of our model. Please let us know if it is clearer now. We would appreciate any other suggestions that can make it better for the final version.
>
> **W5:** We understand the reviewer’s concern. We emphasize that we introduce two new acronyms (FLEX and SCLN) in our paper and the rest of them are common names of tasks, datasets and metrics in the image restoration field. We define all acronyms at first use in main text and for the names of the dataset and metrics we have created Table 13. Now all the acronyms are linked to this table and readers can jump to it using a hyperlink. Spelling out full forms of all these acronyms every time will significantly increase the length of the paper.
>
> **W6:** We acknowledge this and have now expanded the captions to be more detailed wherever necessary. The yellow, cyan and red boxes indicate critical areas to compare between methods. We had previously provided the explanation of the boxes in the main text but have now added it in the caption too. We have also expanded the boxes and isolated them on top of the figure so it is easier to assess the visual quality of the images. Thank you for this suggestion.

---

> ### Author Response · Authors · 2025-11-24
> **All weaknesses and questions addressed point by point. Q1-5**
>
> **Q1:** Thank you for this question. We have now reported teacher-only results in Table 2. The student outperforms the teacher because the teacher is trained with standard reconstruction losses only. The student model learns from the teacher's high-quality features plus velocity field regularization and feature loss alignment, which acts as additional supervision. Rectified flow provides a generative smoothing effect that reduces overfitting to training artifacts.
>
> **Q2:** Yes, the teacher uses identical U-Net Transformer architecture for all restoration tasks. Only the training data changes.
>
> **Q3:** Our teacher is not borrowed from prior work, it is a novel design with: **SCLN (ours):** Spatial-channel layer normalization (no prior work uses this). **QK normalization (ours):** Stabilized attention for degraded inputs (adapted from LLM literature but new to restoration). As for the losses, we include auxiliary constraints along with well established Retinex priors which further improve performance as seen in Figure 3. The Reti-Diff teacher model uses standard Layer Normalization without QK norm. We have now added Table 1 which shows SCLN adds only 0.33% overhead in FP32 while improving training stability as seen in Figure 3.
>
> **Q4:** The student uses the same U-Net Transformer backbone as the teacher, with two novel additions: **Velocity predictor networks (ours):** Residual MLPs that learn rectified flow dynamics (rectified flow theory is well established but never used by any prior method for image restoration), **FLEX loss (ours):** Novel feature alignment knowledge distillation strategy. This loss builds on previous work which is cited in related works.
>
> The same backbone architecture enables direct feature-space distillation. Using a different student architecture (e.g., CNN-based) would theoretically be possible, but will add complexity without clear benefit. Our main goal is to demonstrate that generative distillation improves over static distillation (Reti-Diff), not to compress to a smaller model, though our method could theoretically enable that. We will explore this in future work, as it is out of scope for this project.
>
> **Q5:** Yes it is, confirming the generality of our approach. We use identical student/teacher architecture across all experiments, and the same hyperparameters (λ_FLEX=0.15, τ_SNR=0.4, percentile=95%). For real-world image restoration, we test on five unpaired datasets: DCIM, LIME, MEF, NPE, and VV. Here we use the LOL-v2-syn pretrained model for inference since these datasets have no ground truth.
>
> **We also validate transfer learning results by our generalization experiments in Table 8:** models trained on LOL-v1 transfer to LOL-v2-real with 22.27 PSNR (only -0.70 dB drop from in-domain 22.97), demonstrating robust cross-dataset performance without architecture modification.
>
> **Understanding your concern for further validation we have now added results on ExDark low light object detection task in Table 10.** Here low light images from the ExDark dataset were restored and an object-detection task was performed using the YOLOv3 model. It shows our restored images achieve 77.1% mAP vs. 75.8% for Reti-Diff, confirming that restoration quality translates to real-world vision tasks. **Table 11 has also been added**, which shows comparison against previous image restoration transformer models. **We have also added Figure 9**, which shows failure modes on the hardest dataset SICE-Mix, where all models consistently perform the worst. Extreme overexposure/underexposure with complete information loss.

---

### Official Review · Reviewer_Dxbb · 2025-11-01

**Soundness:** 4
**Presentation:** 4
**Contribution:** 3
**Rating:** 4
**Confidence:** 4

**Summary:**

The paper studies degraded image restoration through knowledge distillation framed as a generative rectified-flow process from teacher to student. A teacher transformer is trained with physics-inspired auxiliaries (e.g., Retinex, anisotropic diffusion, polarized HVI color) and attention stabilizers (SCLN, QK-norm). The student learns velocity fields along straight-line feature paths and uses a FLEX loss combining cross-normalization, percentile masking, and resolution weighting. The claimed benefits are few-step inference and competitive quality across multiple restoration tasks.

**Strengths:**

1. A cohesive rectified-flow perspective on feature-level distillation that is implementable with simple Euler updates and clear path parameterization.
2. A concrete, reproducible teacher recipe that mixes classical priors with modern transformers and documents key loss weights and stabilization tricks.
3. A two-stage student training pipeline that is logically structured and easy to follow.
4. The FLEX loss addresses teacher–student distribution mismatch in a targeted way via student-statistic normalization and percentile outlier handling.
5. The few-step inference angle is practical for deployment scenarios where latency is constrained.

**Weaknesses:**

1. The teacher pipeline appears to use ground-truth signals during pretraining or feature extraction, risking supervision leakage and motivating an LQ-only teacher-target variant.
2. Student-side ablations are shallow because the effects of rectified flow, FLEX, trajectory consistency, and step-size scheduling are not disentangled and there is no sensitivity to percentile, SNR, or step count.
3. Positioning against modern transformer-aware KD remains largely narrative without controlled head-to-head experiments on attention, token, or logit distribution alignment.
4. Evidence across datasets and metrics is uneven, with limited discussion of fidelity–perceptual trade-offs and missing analysis of clear failure cases.
5. The proposed method needs to be tested on other image restoration tasks (such as super-resolution and denoising).

**Questions:**

1. Are any KD targets derived from inputs containing ground truth (e.g., concatenated [LQ, GT] or GT-based Retinex)? If so, what is the performance when targets are computed from LQ-only inputs?
2. Can you add per-component ablations: rectified flow on/off, FLEX on/off, and within FLEX: cross-normalization only vs. +percentile vs. +resolution weighting, with sensitivity to the percentile threshold and SNR/step-size settings?
3. How does the method compare to attention-/token-level transformer KD under identical student backbones and training budgets?
4. Can you expand results across all stated restoration categories with both fidelity (PSNR/SSIM) and perceptual (LPIPS/CLIP-IQA/FID) metrics, and discuss cases where metrics disagree?
5. Will you release code, training/evaluation scripts, and the exact list of selected inference steps and hyperparameters to ensure full reproducibility?

---

> ### Author Response · Authors · 2025-11-24
> **All weaknesses and questions addressed point by point. W1-3**
>
> **We thank the reviewer for their critique. We address each comment in this rebuttal. The paper now has new Tables with all the mentioned required experiments the reviewer requested. The Figures are now updated too. The updated text is in blue. We believe our updated manuscript has substantial technical and applied contributions in the field of image restoration. We would kindly request the reviewer to reassess our paper in view of this rebuttal. If the reviewer requires any further experiments we are happy to discuss and perform them for the final version of the paper.**
>
> ---
>
> **W1:** We would like to clarify: The architecture does not leak GT information during inference. The concatenation [LQ; GT] occurs only during training for the Retinex and Image encoders to learn robust mappings of feature representations. At inference time: Teacher inference: Only LQ is processed. The encoders use LQ features exclusively. Student inference: The student receives only LQ inputs and generates features via rectified flow without any GT dependency.
>
> The dual-input with LQ and GT paired image training design follows standard practice in generative restoration modelling (Reti-Diff, DiffIR) to learn feature extractors that capture the mapping from degraded to clean distributions. Previous models have extensively explored ablations of dual input modelling and we do not claim any novelty on this part. Hence we have skipped ablations of single input formulations for the sake of brevity. If the reviewer still wishes we can easily train a model with this ablation for the final version.
>
> **W2:** We acknowledge this and have now addressed this concern comprehensively in the revised manuscript with Table 9 (Student Ablation Study), which fully disentangles every component. The ablation progression shows: (1) RF without TC or FLEX (24.10 PSNR) underperforms baseline static KD (25.35), demonstrating that rectified flow alone is insufficient; (2) Adding trajectory consistency (TC) recovers performance to 24.27 PSNR while reducing BIQI from 17.95 to 9.02, showing TC's role in stabilizing feature generation; (3) FLEX cross-normalization (CN) jumps to 25.64 PSNR (+1.37 dB), confirming that distribution alignment is the primary driver; (4) Adding percentile masking (PM) provides the largest single gain to 27.82 PSNR (+2.18 dB), indicating that outlier handling is essential for training stability; and (5) Resolution weighting (RW) adds a marginal 0.02 dB but improves convergence consistency.
>
> For hyperparameter sensitivity analysis p=95% provides optimal balance: lower values (85-90%) are too aggressive and remove useful information, while 99% retains outliers that destabilize training. SNR threshold shows robust performance in [0.4, 0.6] range.
> Please note that these experiments are run on a single seed and no statistical testing can be done on them. We will run these ablations on 3 random seeds and perform hypothesis tests to ensure the gains are statistically valid in the camera read version due to computation resources being constrained during the rebuttal period.
>
> **W3:** Direct comparison with attention-/token-level transformer KD (DeiT, TinyBERT) is fundamentally mismatched for several reasons.
> 1. **Different objectives:** These methods transfer classification knowledge (logits, attention maps), while we transfer generative feature distributions for pixel-level restoration.
> 2. **Architecture mismatch:** Our teacher uses Retinex-conditioned attention with SCLN, which has no equivalent in standard vision transformers. Attention map alignment would require identical Q/K/V dimensions and head configurations.
> 3. **Existing baseline:** Reti-Diff (ICLR'25) represents the state-of-the-art transformer KD for restoration, using static feature matching. Our Table 9 shows +2.49 PSNR over Reti-Diff's KD approach (27.84 vs 25.35), demonstrating superiority within the restoration paradigm. We emphasize that our contribution is in reformulating KD as a generative process, not competing with classification-domain KD methods. The fair comparison is against Reti-Diff, which we consistently outperform.

---

> ### Author Response · Authors · 2025-11-24
> **All weaknesses and questions addressed point by point. W4-5, Q1-5.**
>
> **W4:** We acknowledge this, we previously only provided metrics that previous papers reported their numbers on for fair comparison. **We have now added Table 12** with 15 datasets × 10 metrics, including both fidelity (PSNR/SSIM) and perceptual (FID/LPIPS/NIQE/BRISQUE) measures. Key observations: LOL-v1: High PSNR (27.85) + Low FID (38.67),  good fidelity-perceptual alignment. SICE-mix: Lower PSNR (15.04) but acceptable LPIPS (0.39),  perceptual quality maintained despite pixel-level noise. BAID: Best FID (15.83) across all datasets, strong perceptual realism.
>
> **We have also added Figure 9**, which shows failure modes on the hardest dataset SICE-Mix, where all models consistently perform the worst. Extreme overexposure/underexposure with complete information loss. Artifacts: Color bleeding, checkerboard patterns, hallucinatory textures. Root cause: When SNR falls below threshold, velocity predictors struggle to generate meaningful features, relying on learned priors that introduce unrealistic reconstructions. We will explore how to resolve these issues in the future. Trade-off discussion: In degraded regions (SICE failure cases), PSNR drops but LPIPS remains reasonable because the model generates plausible textures, even if not pixel-accurate. This highlights the generative nature of our approach as it fills missing information rather than strictly reconstructing.
>
> **W5:** We acknowledge your request. Super-resolution and denoising tasks are a separate field in computer vision that have their own 5-10 other reference datasets. Our method is designed for illumination-degraded image restoration (low-light, underwater, backlit), where Retinex priors and color spaces are physically motivated. Extension to SR/denoising would require: Replacing Retinex and color decomposition with noise/blur-specific priors. Retraining encoders for different degradation types. And comparing against completely different baseline methods and papers.
>
> This represents significant additional work beyond the scope of this paper. However, the FLEX loss and rectified flow framework are task-agnostic and could transfer to these domains, we propose this as high-impact future work. **If the reviewer still wishes, we can add results on one dataset for denoising and super resolution, however we feel it is out of scope for this project.**
>
> **Understanding your concern for further validation we have now added results on ExDark low light object detection task in Table 10.** Here low light images from the ExDark dataset were restored and an object-detection task was performed using the YOLOv3 model. It shows our restored images achieve 77.1% mAP vs. 75.8% for Reti-Diff, confirming that restoration quality translates to real-world vision tasks. **Table 11 has also been added**, which shows comparison against previous image restoration transformer models.
>
> ---
>
> **Q1:** Addressed in W1 please let us know if that answers your question.
>
> **Q2:** Yes **we have now provided sensitivity analysis in Table 9**. Please let us know if any more analysis is needed we are happy to perform more experiments for the final version.
>
> **Q3:** Addressed in W3 please let us know if that answers your question.
>
> **Q4:** Addressed in W4 please let us know if that answers your question.
>
> **Q5:** Yes, absolutely. We have the code ready and will release: Complete training/evaluation scripts. Pre-trained teacher and student weights. Fully configurable inference scripts. Our github is privatized currently for the double blind review process and we will make it public upon acceptance.

---

### Author Response · Authors · 2025-11-25
**Updated PDF and ZIP content.**

Dear Reviewers and AC,

Please find the updated PDF file and ZIP file. The updated PDF has all the requested revisions in blue, and the ZIP has highest quality version of the figures since the PDF only allows 50mb size.

If more experiments or clarifications are required please let us know during the discussion period.

Thank you!

Authors

---

### Note · Authors · 2026-01-26

I have read and agree with the venue's withdrawal policy on behalf of myself and my co-authors.

---

### Meta-Review · Area_Chair_5B3M · 2026-01-07

**Summary:**

This paper received mixed initial review comments. One of the reviewers show positive feedbacks after the rebuttal, and three of the reviewers rated "borderline reject" or "reject" before the rebuttal.

AC read the paper, review comments and the detailed rebuttal provided by the authors and appreciated the idea and effort of exploring generative distillation on top of a regression-based model, and the authors well demonstrated extensive ablation studies in the rebuttal to show the effectiveness of each components (loss, tricks) to further boost the metrics. Overall the complexity and efforts of the paper delivered valuable messages.

While some concerns are still outstanding,

- The scope of the paper is mainly "illumination-degraded image restoration" as mentioned by the authors, while the title and the introduction formulates it as a general image restoration approach. It's strongly suggested to narrow down the scope if the proposed method cannot well support denoising and super-resolution.
- the training strategy is a bit over-engineering, and over-complicated training phases will cause reproducing issues.
- the paper lacks in core contributions, while stacking multiple existing or revised engineering tricks (RF, SCLN, FLEX). However, a better presentation might resolve this issues, like emphasizing the generative distillation parts, and downweight other loss revision and attention norm design.
- some visual results in the paper contain highly compressed artifacts (lower-res and lower-quality), like in Figure 5. Showing low-quality results in restoration paper is not good enough.

These points make the paper borderline.

However, the major concern from the AC is the claim that the proposed RestoRect especially the FLEX is designed for heterogeneous teacher-student KD, while the experiments are using identical architecture between teacher and student. We understand the main contribution of the paper is the generative distillation, while model compression and architecture change are more important for practical values. Also the authors did not provide ablations on the traditional KD, and the results of RetiDiff in Table 9 is a bit misleading.

Given the above reasons, AC does not recommend an acceptance. To further improve the paper, AC suggests (1) narrow down the scope and claim to "illumination-based restoration" only. (2) conduct experiments on heterogeneous teacher-student. It will make the paper stronger.

**Reviewer Concerns:**

- Reviewer Dxbb: the authors well explained the misunderstanding of the training-inference leakage and made up many student-side ablation to address the concerns. Also the authors clarified the scope of the paper, and added more results plus failure cases. All the concerns from this reviewer should have been addressed well.
- Reviewer wXE6: the review comments focus more on the presentation of the paper, but not technical details. AC did not weight heavily on this review. The authors clarified the details of the student / teacher regarding architecture and the performance. Overall presentation styles are not changed, so the reviewer may still have remaining concerns.
- Reviewer jwUY: AC believes the authors gave good responses to the novelty concerns from this reviewer. The additional computational cost and profiling make the design more convincing. All the concerns of this reviewer is properly addressed.
- Reviewer MwNg: All concerns resolved.

**Reviewer Scores:**

- Reviewer Dxbb: the scores might be increased.
- Reviewer wXE6: it is not that possible to increase the scores.
- Reviewer jwUY:  it is possible the scores will become less negative, but harder for the reviewer to make it positive.
- Reviewer MwNg: All concerns resolved, so the score will remain positive.

---

### Decision · Program_Chairs · 2026-01-26

Reject